# Identification and biological characteristics of *Fusarium tobaccum* sp. nov., a novel species causing tobacco root rot in Jilin Province, China

Zhao Xie,[1] Chong Gao,[2] Xiaoyan Zhang,[1] Mengzhu Du,[1] Jun Wang,[1] Xue Wang,[1,3] Baohui Lu,[1,3] Changqing Chen,[1,3] Lina Yang,[1,3] Yanjing Zhang,[1,3] Jie Gao[1,3]

**ABSTRACT**    Fusarium wilt of tobacco (FWT), caused by *Fusarium* spp., has emerged as a severe threat to tobacco production in China. In all, 132 isolates of *Fusarium* were isolated from tobacco and pathogenic to tobacco-causing FWT in Jilin Province, China. In this study, we identified 7 of 132 isolates as a novel species *Fusarium tobaccum* sp. nov. Zhao Xie & Jie Gao, using multi-gene phylogenetic analyses of translation elongation factor (*tef1*), β-tubulin (*tub2*), calmodulin (*cmdA*), and RNA polymerase II second largest subunit (*rpb2*) genes, along with subtle morphological differences. Isolates of *F. tobaccum* sp. nov. were clustered in a distinct branch in the maximum parsimony phylogenetic tree generated from the sequences of *tef1-rpb2-tub2-cmdA* and can be distinguished from closely related species *F. cugenangense*, *F. callistephi*, and *F. elaeidis*. The morphological characteristics of *F. tobaccum* sp. nov. are distinct from other *Fusarium* species. *F. tobaccum* sp. nov. exhibits abundant aerial mycelia and pigment production on potato dextrose agar (PDA), microconidia with 0–1 septa, and macroconidia with 2–5 septa on carnation leaf-piece agar (CLA), and produces abundant chlamydospores on Spezieller Nährstoffarmer agar (SNA) and CLA. The mycelia of *F. tobaccum* exhibited optimal growth at a pH of 7.1 and a temperature of 23.6℃. Sucrose and $NaNO_3$ significantly promoted the mycelial growth of *F. tobaccum*. PD medium was optimal for total sporulation. However, the sporulation ratios of the macrospores of *F. tobaccum* in PD, SN, and CMC were relatively low (0.48%, 2.51%, and 2.16%, respectively). These findings provided valuable insights into the morphological and biological characteristics of *F. tobaccum*.

**IMPORTANCE**    *Fusarium* wilt of tobacco (FWT) is a prevalent issue in tobacco-growing regions globally, leading to significant losses in yield and quality. This study identified *F. tobaccum* sp. nov., a novel species of *Fusarium* causing FWT in China. The identification was based on multi-gene phylogenetic analyses and morphological characteristics. The effects of temperature, pH, carbon source, nitrogen source, medium, and light on the mycelial growth of *F. tobaccum* sp. nov. were determined. These findings might contribute to future research on the pathogenic mechanisms of this novel species and the development of strategies to control FWT.

**KEYWORDS**    biological characteristics, *Fusarium tobaccum* sp. nov, morphology, wilt, tobacco

**Peer Reviewers** Laith Khalil Tawfeeq Al-Ani, University of Baghdad, Baghdad, Iraq; Chunwei Wang, Shanxi Agricultural University, Jinzhong, China

Address correspondence to Jie Gao, jiegao115@126.com, or Yanjing Zhang, yjzhang77@126.com.

The authors declare no conflict of interest.

See the funding table on p. 19.

Tobacco (*Nicotiana tabacum* L.) is an economically important crop in many countries, including China. However, diseases and pests significantly reduce tobacco yield and quality. One of the major diseases affecting tobacco production is *Fusarium* wilt of tobacco (FWT), a soilborne fungal disease caused by *Fusarium* spp. FWT results in diseased plants lacking commercial value, and hence has become a significant threat to

tobacco production. The disease was first reported in the United States and has since been reported in various countries, including South Korea, Argentina, Zimbabwe, Italy, Greece, and China (1–7). In China, FWT was first reported in Guizhou Province; it has since been found in several other provinces, such as Yunnan, Henan, Hunan, Shandong, Fujian, and Hubei (8–14). The incidence and severity of FWT have increased in recent years, with annual incidences ranging from 10% to 30%; incidences as high as 100% have been observed in severe cases.

*Fusarium* is one of the most economically destructive soilborne plant pathogens, comprising at least 300 phylogenetically distinct species/species complexes (15). *Fusarium oxysporum*, which is often referred to as the *F. oxysporum* species complex (FOSC), comprises a group of fungi causing root rot and vascular wilt diseases in economically important crops globally. The plant pathogenic isolates of FOSC are further classified into different *formae speciales* (f. sp.) and races based on their host specificity (16, 17). The causal agent of FWT was first identified as *F. oxysporum* f. sp. *nicotianae* (4, 6). To date, 17 *Fusarium* spp. causing wilt of tobacco have been reported globally, including *F. brachygibbosum*, *F. chlamydosporum*, *F. commune*, *F. equiseti*, *F. falciforme*, *F. fujikuroi*, *F. kyushuense*, *F. meridionale*, *F. oxysporum*, *F. proliferatum*, *F. redolens*, *F. scirpi*, *F. semitectum*, *F. sinensis*, *F. solani*, *F. tricinctum*, and *F. verticillioides* (5, 12–14, 18–28). *F. oxysporum* and *F. solani* are the dominant species causing FWT.

In recent years, FWT has become a prevalent root disease in most tobacco-planting areas of Jilin Province, posing a significant threat to tobacco production. No external symptoms of infection are apparent in the early stages of FWT. However, as the disease progresses, the lower to middle leaves of tobacco plants gradually become yellow and wilt, leading to slow growth and extensive wilting in diseased plants. Severe cases can even result in plant death, causing significant losses in commercial value (29, 30). Previous studies in our laboratory identified three *Fusarium* species complexes capable of causing FWT in Jilin Province, China, including *Fusarium oxysporum* species complex (FOSC), *F. solani* species complex, and *F. nisikadoi* species complex; 320 FOSC isolates were obtained, accounting for 83% of the total isolates (unpublished data).

In this study, a novel species *Fusarium tobaccum* sp. nov. was found belonging to the FOSC during a survey of the FWT conducted from 2018 to 2020 in Jilin Province, China. This pathogen had not previously been reported causing FWT. Therefore, we aimed to identify this causal agent, based on the analyses of multiple gene sequences and morphological characteristics. In addition, we extensively characterized the biological features of *Fusarium tobaccum* sp. nov. to gain a comprehensive understanding of its properties.

## MATERIALS AND METHODS

### Fungal isolation and preservation

From 2018 to 2020, FWT-like diseased samples were collected in five towns in four counties or cities in Jilin Province, China. A total of 132 pure cultures were obtained and the cultures were stored using the sterile filter paper method in a refrigerator at –20°C and activated on potato dextrose agar (PDA) plates before use. Then seven isolates were selected for further study (Table 1). *F. cugenangense* LH41 was selected as the control isolate.

### Media used in the experiment

PDA medium (potato 200 g, glucose 15 g, agar 20 g, and distilled water to a volume of 1,000 mL) was used for the preservation, activation, and morphological observations of fungi. Potato dextrose (PD) medium (potato 200 g, glucose 15 g, and distilled water to a volume of 1,000 mL) was used for sporulation tests. Potato saccharose agar (PSA) medium (potato 200 g, sucrose 15 g, agar 20 g, and distilled water to a volume of 1,000 mL) was used to evaluate biological characteristics. Potato carrot agar

**TABLE 1** Origin and pathogenicity of *Fusarium tobaccum* sp. nov. isolates from tobacco included in this study[a]

| Isolate | Origin | Pathogenicity |
|---|---|---|
| LH532 | Tuoyaoling town, Liuhe County, Jilin province, China (E125°57′28.706″,N42°10′33.456″） | ++ |
| LH912 | Tuoyaoling town, Liuhe County, Jilin province, China (E125°57′28.706″,N42°10′33.456″） | ++ |
| LH1294 | Tuoyaoling town, Liuhe County, Jilin province, China (E125°57′28.706″,N42°10′33.456″） | +++ |
| LH1297 | Tuoyaoling town, Liuhe County, Jilin province, China (E125°57′28.706″,N42°10′33.456″） | +++ |
| LH1794 | Tuoyaoling town, Liuhe County, Jilin province, China (E125°57′28.706″,N42°10′33.456″） | + |
| LH1904 | Tuoyaoling town, Liuhe County, Jilin province, China (E125°57′28.706″,N42°10′33.456″） | ++ |
| LH156101 | Tuoyaoling town, Liuhe County, Jilin province, China (E125°57′28.706″,N42°10′33.456″） | +++ |

[a] +, ++, and +++ represent disease grade of 1-2, 3-4, and 5, respectively.

(PCA) medium (potato 20 g, carrot 25 g, agar 20 g, and distilled water to a volume of 1,000 mL) was used to evaluate biological characteristics. Carnation leaf-piece agar (CLA) medium (agar 20 g, distilled water to a volume of 1,000 mL, and carnation leaves) was used to observe the morphological characteristics of fungi. Sterile carnation leaves (3–5 mm$^2$) were placed on the surface of a sterilized water agar (WA) plate cooled to 45°C. Next, 10–12 pieces of the leaves were added to a 9-cm-diameter culture dish, and the fungal mycelial plugs appeared close to the carnation leaves. Spezieller Nährstoffarmer agar (SNA) medium (KCl 0.5 g, MgSO$_4$·7H$_2$O 0.5 g, KNO$_3$ 1 g, KH$_2$PO$_4$ 1 g, sucrose 0.2 g, glucose 0.2 g, agar 20 g, and distilled water 1,000 mL) was used for the morphological observations of fungi. Spezieller Nährstoffarmer (SN) liquid medium (KCl 0.5 g, MgSO$_4$·7H$_2$O 0.5 g, KNO$_3$ 1 g, KH$_2$PO$_4$ 1 g, sucrose 0.2 g, glucose 0.2 g, and distilled water 1,000 mL) was used for fungal sporulation. Sodium carboxymethyl cellulose (CMC) medium (sodium carboxymethyl cellulose 15 g, NH$_4$NO$_3$ 1 g, yeast extract 1 g, MgSO$_4$·7H$_2$O 0.5 g, KH$_2$PO$_4$ 1 g, and distilled water 1,000 mL) was used for fungal sporulation. Czapek (Czapek–Dox) medium (sodium nitrate 3 g, dipotassium hydrogen phosphate 1 g, magnesium sulfate 0.5 g, potassium chloride 0.5 g, ferrous sulfate 0.01 g, sucrose 30 g, agar 20 g, and distilled water 1,000 mL), oatmeal agar (OA) medium (oatmeal 60 g, agar 12.5 g, and distilled water 1,000 mL), alkyl ester agar (AEA) medium (yeast powder 5 g, NaNO$_3$ 6 g, KH$_2$PO$_4$ 1.5 g, KCl 0.5 g, MgSO$_4$ 0.25 g, glycerol 20 mL, agar 15 g, and distilled water 1,000 mL), and V8 juice agar medium (V8 juice 200 mL, CaCO$_3$ 3 g, agar 20 g, and distilled water 1,000 mL) (31–36) were used to evaluate biological characteristics.

## DNA isolation, polymerase chain reaction amplification, and sequencing

Total genomic DNA was extracted from seven isolates (LH532, LH912, LH1294, LH1297, LH156101, LH1794, and LH1904) cultured on PDA for 7 days at 25°C under continuous light using the CTAB (cetyl-methyl-ammoniumbromide) method (37). Partial gene sequences of seven isolates were determined for β-tubulin (*tub2*), calmodulin (*cmdA*), RNA polymerase II second largest subunit (*rpb2*), and translation elongation factor 1 (*tef1*) using previously described polymerase chain reaction (PCR) protocols (38–42) . The primer pairs T1/CYLTUB1R (43, 44) for *tub2*, Cal228F/CAL2Rd (45, 46) for *cmdA*, 5f2/7cr (47, 48) for *rpb2*, and EF1/EF2 (38) for *tef1* were used for PCR amplification (Table 2). The integrity of the sequences was ensured by sequencing the amplicons in both directions using the same primer pairs. The consensus sequences for each locus were assembled in MEGA (Molecular Evolutionary Genetics Analysis) v.7 (49). All sequences were uploaded to GenBank to obtain the GenBank accession numbers (Table 3).

**TABLE 2** Primers, sequences, and their sources used in this study

| Genes | Primer | Sequence | References |
|---|---|---|---|
| *tub2* (β-tubulin) | T1 | AACATGCGTCATTGTAAGT | O'Donnell and Cigelnik |
| | T2 | TAGTGACCCTTGGCCCAGTTG | (43), Crous et al. (44) |
| *tef1* (translation elongation factor 1α) | EF1728F | CATCGAGAACCAGAAGG | O'Donnell et al. (38) |
| | EF2 | GGA(G/A)GTACCAGT(G/C)ATCATCTT | |
| *cmdA* (calmodulin) | CAL228F | GAGTTCAAGGAGGCCTTCTCCC | Carbone and Kohn (45), |
| | CAL2Rd | TGRTCNGCCTCDCGGATCATCTC | Groenewald et al. (46) |
| *rpb2* (RNA polymerase II second largest subunit) | RPB25F2 | GGGGWGAYCAGAAGAAGGC | Liu et al. (47), Sung et al. |
| | RPB27cR | CCCATRGCTTGYTTRCCCAT | (48) |

## Phylogenetic analyses

Maximum parsimony was used for phylogenetic analyses conducted using the individual loci and a multi-locus sequence data set including the *cmdA*, *rpb2*, *tef1*, and *tub2* sequences. The analyses were conducted using Phylogenetic Analysis Using Parsimony (50), with phylogenetic relationships estimated using heuristic searches with 1,000 randomly added sequences. The tree-bisection-reconnection algorithm was used, with the branch swapping option set to "best trees" only. All characters were weighted equally, and alignment gaps were treated as a fifth character state. The measures calculated for parsimony included tree length (TL), consistency index (CI), retention index (RI), rescaled consistency index (RC), and homoplasy index (HI). Bootstrap (BS) analyses (51) were conducted using 1,000 bootstrap replicates.

## Morphological characteristics

The morphology of a representative isolate of *F. tobaccum* sp. nov. LH156101 was characterized following the protocols described by Leslie and Summerell (32) and Lombard et al. (42). The colony morphology, growth rate, texture, odor, and pigment production were evaluated on the PDA medium. The morphology of macroconidia and microconidia ($n$ = 100) was evaluated on SNA medium, and the morphology of the sporulation structure and chlamydospore on CLA medium was measured after 7 days at 25°C under a 12-h light/12-h dark photoperiod.

## Assessment of pathogenicity

The pathogenicity of all seven isolates on tobacco was assessed to validate Koch's postulates. The experiment was conducted in a greenhouse at Jilin Agricultural University.

### Preparation of tobacco seedlings

The soil was sterilized at 160°C for 1 h and placed in plates (60 × 30 cm$^2$) until two-thirds of the total volume was filled. It was then sprayed thoroughly with water. Tobacco seeds (cv. Jiyan 9) were sterilized with 0.1% AgNO$_3$ for 15 min, sown, and then covered with sterile soil. When the tobacco seedlings had two leaves, they were moved to a plate containing 100 pots with sterile soil and one plant per pot. All tobacco seedlings with four leaves were prepared for inoculation at 25°C under a 12-h light/12-h dark photoperiod.

### Inoculum preparation

Fresh mycelial plugs were taken from the edge of colonies incubated on PDA medium at 25°C for 7 days and inoculated into 100 g of sterilized barley grains in a conical flask (250 mL) for 7 days at 28°C to prepare the barley grain–fungus culture for inoculation (52).

**TABLE 3** GenBank accession numbers of isolates included in this study (newly generated sequences are indicated in bold)

| Species[a] | Isolate | GenBank accession number | | | | Species | Isolate | GenBank accession number | | | |
|---|---|---|---|---|---|---|---|---|---|---|---|
| | | tef1 | tub2 | rpb2 | cmdA | | | tef1 | tub2 | rpb2 | cmdA |
| Fusarium callistephi | CBS 115423 | MH484996 | MH485087 | MH484905 | MH484723 | F. nirenbergiae | CBS 196.87 | MH484977 | MH485068 | MH484886 | MH484704 |
| F. callistephi | CBS 187.53 | MH484966 | MH485057 | MH484875 | MH484693 | F. nirenbergiae | CBS 744.79 | MH484973 | MH485064 | MH484882 | MH484700 |
| F. carminascens | CPC 25792 | MH485025 | MH485116 | MH484934 | MH484752 | F. nirenbergiae | CBS 758.68 | MH484968 | MH485059 | MH484877 | MH484695 |
| F. carminascens | CPC 25793 | MH485026 | MH485117 | MH484935 | MH484753 | F. nirenbergiae | CBS 840.88 | MH484978 | MH485069 | MH484887 | MH484705 |
| F. carminascens | CPC 25795 | MH485027 | MH485118 | MH484936 | MH484754 | F. nirenbergiae | CBS 115416 | MH484993 | MH485084 | MH484902 | MH484720 |
| F. carminascens | CPC 25800 | MH485028 | MH485119 | MH484937 | MH484755 | F. nirenbergiae | CBS 115417 | MH484994 | MH485085 | MH484903 | MH484721 |
| F. contaminatum | CBS 111552 | MH484991 | MH485082 | MH484900 | MH484718 | F. nirenbergiae | CBS 115419 | MH484995 | MH485086 | MH484904 | MH484722 |
| F. contaminatum | CBS 114899 | MH484992 | MH485083 | MH484901 | MH484719 | F. nirenbergiae | CBS 115424 | MH484997 | MH485088 | MH484906 | MH484724 |
| F. contaminatum | CBS 117461 | MH485002 | MH485093 | MH484911 | MH484729 | F. nirenbergiae | CBS 123062 | MH485010 | MH485101 | MH484919 | MH484737 |
| F. cugenangense | CBS 620.72 | MH484970 | MH485061 | MH484879 | MH484697 | F. nirenbergiae | CBS 130300 | MH485016 | MH485107 | MH484925 | MH484743 |
| F. cugenangense | CBS 130308 | MH485011 | MH485102 | MH484920 | MH484738 | F. nirenbergiae | CBS 130301 | MH485017 | MH485108 | MH484926 | MH484744 |
| F. cugenangense | CBS 130304 | MH485012 | MH485103 | MH484921 | MH484739 | F. nirenbergiae | CBS 130303 | MH485014 | MH485105 | MH484923 | MH484741 |
| F. cugenangense | CBS 131393 | MH485019 | MH485110 | MH484928 | MH484746 | F. nirenbergiae | CPC 30807 | MH485041 | MH485132 | MH484950 | MH484768 |
| F. curvatum | CBS 141.95 | MH484985 | MH485076 | MH484894 | MH484712 | F. odoratissimum | CBS 794.70 | MH484969 | MH485060 | MH484878 | MH484696 |
| F. curvatum | CBS 238.94 | MH484984 | MH485075 | MH484893 | MH484711 | F. odoratissimum | CBS 102030 | MH484989 | MH485080 | MH484898 | MH484716 |
| F. curvatum | CBS 247.61 | MH484967 | MH485058 | MH484876 | MH484694 | F. odoratissimum | CBS 130310 | MH485013 | MH485104 | MH484922 | MH484740 |
| F. duoseptatum | CBS 102026 | MH484987 | MH485078 | MH484896 | MH484714 | F. oxysporum | CBS 221.49 | MH484963 | MH485054 | MH484872 | MH484690 |
| F. elaeidis | CBS 217.49 | MH484961 | MH485052 | MH484870 | MH484688 | F. oxysporum | CBS 144134 | MH485044 | MH485135 | MH484953 | MH484771 |
| F. elaeidis | CBS 218.49 | MH484962 | MH485053 | MH484871 | MH484689 | F. oxysporum | CBS 144135 | MH485045 | MH485136 | MH484954 | MH484772 |
| F. elaeidis | CBS 255.52 | MH484965 | MH485056 | MH484874 | MH484692 | F. oxysporum | CPC 25822 | MH485034 | MH485125 | MH484943 | MH484761 |
| F. fabacearum | CPC 25802 | MH485030 | MH485121 | MH484939 | MH484757 | F. pharetrum | CPC 30822 | MH485042 | MH485133 | MH484951 | MH484769 |
| F. fabacearum | CPC 25803 | MH485031 | MH485122 | MH484940 | MH484758 | F. pharetrum | CPC 30824 | MH485043 | MH485134 | MH484952 | MH484770 |
| F. foetens | CBS 120665 | MH485009 | MH485100 | MH484918 | MH484736 | F. libertatis | CPC 28465 | MH485035 | MH485126 | MH484944 | MH484762 |
| F. glycines | CBS 176.33 | MH484959 | MH485050 | MH484868 | MH484686 | F. tardichlamydosporum | CBS 102028 | MH484988 | MH485079 | MH484897 | MH484715 |
| F. glycines | CBS 200.89 | MH484979 | MH485070 | MH484888 | MH484706 | **F. tobaccum sp. nov.** | **LH532** | **OM162136** | **OL437357** | **OL437329** | **OL437385** |
| F. glycines | CBS 214.49 | MH484960 | MH485051 | MH484869 | MH484687 | **F. tobaccum sp. nov.** | **LH912** | **OM162140** | **OL437361** | **OL437333** | **OL437389** |
| F. glycines | CPC 25804 | MH485032 | MH485123 | MH484941 | MH484759 | **F. tobaccum sp. nov.** | **LH1294** | **OM162144** | **OL437365** | **OL437337** | **OL437393** |
| F. glycines | CPC 25808 | MH485033 | MH485124 | MH484942 | MH484760 | **F. tobaccum sp. nov.** | **LH1297** | **MN610673** | **OL437366** | **OL437338** | **OL437394** |
| F. gossypinum | CBS 116611 | MH484998 | MH485089 | MH484907 | MH484725 | **F. tobaccum sp. nov.** | **LH1794** | **OM162148** | **OL437371** | **OL437343** | **OL437399** |
| F. gossypinum | CBS 116612 | MH484999 | MH485090 | MH484908 | MH484726 | **F. tobaccum sp. nov.** | **LH1904** | **OM162150** | **OL437372** | **OL437344** | **OL437400** |
| F. gossypinum | CBS 116613 | MH485000 | MH485091 | MH484909 | MH484727 | **F. tobaccum sp. nov.** | **LH156101** | **MN610680** | **OL437369** | **OL437341** | **OL437397** |
| F. hoodiae | CBS 132474 | MH485020 | MH485111 | MH484929 | MH484747 | F. triseptatum | CBS 258.50 | MH484964 | MH485055 | MH484873 | MH484691 |
| F. hoodiae | CBS 132476 | MH485021 | MH485112 | MH484930 | MH484748 | F. triseptatum | CBS 116619 | MH485001 | MH485092 | MH484910 | MH484728 |
| F. hoodiae | CBS 132477 | MH485022 | MH485113 | MH484931 | MH484749 | F. triseptatum | CBS 119665 | MH485007 | MH485098 | MH484916 | MH484734 |
| F. languescens | CBS 300.91 | MH484982 | MH485073 | MH484891 | MH484709 | F. triseptatum | CBS 130302 | MH485015 | MH485106 | MH484924 | MH484742 |
| F. languescens | CBS 302.91 | MH484983 | MH485074 | MH484892 | MH484710 | F. udum | CBS 177.31 | MH484957 | MH485048 | MH484866 | MH484684 |
| F. languescens | CBS 413.90 | MH484981 | MH485072 | MH484890 | MH484708 | F. veterinarium | CBS 109898 | MH484990 | MH485081 | MH484899 | MH484717 |

**TABLE 3** GenBank accession numbers of isolates included in this study (newly generated sequences are indicated in bold) (*Continued*)

| Species[a] | Isolate | GenBank accession number | | | |
|---|---|---|---|---|---|
| | | tef1 | tub2 | rpb2 | cmdA |
| F. languescens | CBS 645.78 | MH484971 | MH485062 | MH484880 | MH484698 |
| F. languescens | CBS 646.78 | MH484972 | MH485063 | MH484881 | MH484699 |
| F. languescens | CBS 872.95 | MH484986 | MH485077 | MH484895 | MH484713 |
| F. languescens | CBS 119796 | MH485008 | MH485099 | MH484917 | MH484735 |
| F. libertatis | CPC 25782 | MH485023 | MH485114 | MH484932 | MH484750 |
| F. libertatis | CPC 25788 | MH485024 | MH485115 | MH484933 | MH484751 |
| F. libertatis | CPC 28465 | MH485035 | MH485126 | MH484944 | MH484762 |
| F. nirenbergiae | CBS 127.81 | MH484974 | MH485065 | MH484883 | MH484701 |
| F. nirenbergiae | CBS 129.24 | MH484955 | MH485046 | MH484864 | MH484682 |
| F. nirenbergiae | CBS 129.81 | MH484976 | MH485067 | MH484885 | MH484703 |
| F. nirenbergiae | CBS 149.25 | MH484956 | MH485047 | MH484865 | MH484683 |
| F. nirenbergiae | CBS 181.32 | MH484958 | MH485049 | MH484867 | MH484685 |
| F. veterinarium | CBS 117787 | MH485003 | MH485094 | MH484912 | MH484730 |
| F. veterinarium | CBS 117790 | MH485004 | MH485095 | MH484913 | MH484731 |
| F. veterinarium | CBS 117791 | MH485005 | MH485096 | MH484914 | MH484732 |
| F. veterinarium | CBS 117792 | MH485006 | MH485097 | MH484915 | MH484733 |
| F. veterinarium | NRRL54984 | MH485036 | MH485127 | MH484945 | MH484763 |
| F. veterinarium | NRRL54996 | MH485037 | MH485128 | MH484946 | MH484764 |
| F. veterinarium | NRRL62542 | MH485038 | MH485129 | MH484947 | MH484765 |
| F. veterinarium | NRRL62545 | MH485039 | MH485130 | MH484948 | MH484766 |
| F. veterinarium | NRRL62547 | MH485040 | MH485131 | MH484949 | MH484767 |
| Fusarium sp. | CBS 680.89 | MH484980 | MH485071 | MH484889 | MH484707 |
| Fusarium sp. | CBS 128.81 | MH484975 | MH485066 | MH484884 | MH484702 |
| Fusarium sp. | CBS 130323 | MH485018 | MH485109 | MH484927 | MH484745 |

[a]F. tobaccum sp. nov. and their GenBank accession numbers in this study are marked in bold. The others are referenced to Lombard et al. (42).

## Inoculation method

Before inoculation, half of the volume of a pot was filled with sterilized soil, and 5 g of the barley grain–fungus culture was mixed in 5 cm of topsoil in one pot with water; the pot was then incubated at 28°C for 4 days. Healthy tobacco plants in the four-leaf stage were placed in the pot. Each isolate was inoculated on three tobacco seedlings, and three replicates were conducted for each isolate. Tobacco seedlings inoculated with non-fungus barley grain culture were used as a negative control. All the inoculated tobacco seedlings were placed in a greenhouse at 28–30°C and relative humidity exceeding 70% under a 12-h light/12-h dark photoperiod. The fungal isolates were re-isolated from the infected tobacco roots, and the morphological and DNA sequence analyses were conducted to fulfill Koch's postulates (32). The pathogenicity was determined using a disease grade scale 2 weeks after inoculation (53). The stem of each plant was cut diagonally at approximately 1–2 cm above the soil, and the plants were rated using the following scale: 0, no symptoms; 1, slight vascular discoloration with no external symptoms; 2, moderate vascular discoloration with no external symptoms; 3, lower leaves yellowed and with incipient wilting; 4, extensive wilting; and 5, plant death.

## Biological characteristics of *Fusarium tobaccum* sp. nov.

The representative isolate LH156101 of *F. tobaccum* sp. nov. was inoculated on a PDA plate at 25°C for 3 days for activation; it was then transferred onto a new PDA plate or WA plate for further incubation at 25°C for 7 days to test culture and nutrient conditions, as well as sporulation. Three replicates were conducted for each treatment in all experiments; the effects of all factors, including temperature, pH value, light condition, carbon source, nitrogen source, and media, on mycelial growth were determined by measuring colony diameters after 7 days of incubation (54).

### Effects of temperature, pH, and light conditions

Furthermore, 8 mm fungal plugs were taken from the edge of the colony using a hole punch and inoculated in the center of a 90 mm Petri dish with PDA medium in the dark. They were then incubated at different temperatures (5, 10, 15, 20, 25, 30, 35, and 40°C) in an incubator for 7 days. For examining the effect of pH on mycelial growth, PDA media were adjusted with 0.1 M HCl and 0.1 M NaOH to obtain pH values of 4.0, 5.0, 6.0, 7.0, 8.0, 9.0, 10.0, and 11.0. An 8-mm-diameter plug was placed in the center of a 90 mm Petri dish with PDA medium and incubated in the dark for 7 days. For examining the effect of light conditions on mycelial growth, an 8-mm-diameter fungal plug was placed in the center of a 90 mm Petri dish with PDA medium and incubated under full light, a 12-h light/12-h dark photoperiod, and complete darkness for 7 days.

### Effects of carbon and nitrogen sources and different media

Czapek–Dox medium was used as the basic medium for investigating the utilization of nitrogen and carbon sources (33–35). Then, 30 g of sucrose was replaced with 30 g of sucrose, lactose, maltose, trehalose, starch, mannitol, and glycerol to explore the effects of different carbon sources on growth. Furthermore, 3 g of sodium nitrate was replaced with 3 g of sodium nitrate, ammonium nitrate, potassium nitrate, peptone, beef extract, cysteine, and aspartic acid to examine the effects of different nitrogen sources on growth. An 8-mm-diameter WA mycelial plug was transferred to the center of each sole carbon source medium and sole nitrogen source medium (55). PDA, PSA, PCA, SNA, V8, OA, Czapek, and AEA media were used to investigate the effects of different media on mycelial growth. An 8-mm-diameter WA mycelial plug was transferred to the center of each medium and incubated in the dark for 7 days.

### Effects of SN, PD, and CMC liquid media on sporulation and the proportion of macroconidia

Two 8-mm-diameter mycelial plugs were placed in each conical flask (250 mL) at 25℃. The flask was shaken horizontally at 150 rpm and incubated in the dark. The cultures were taken after 1, 1.5, 2, 3, 5, and 7 days of inoculation to examine total sporulation and macroconidia production using a blood cell counting board under an optical microscope (Motic BA210, Fuzhou, China), and the total conidiation and proportion of macroconidia were calculated.

### Statistical analysis

The data were analyzed using IBM SPSS Statistics V22.0 software (IBM Inc., NY, USA). The model that best fits the individual data points was selected, and SPSS Statistics was used to confirm the selected model. The optimal temperature and pH value of the regression curves were calculated based on the regression equations generated using SPSS Statistics. Duncan's test was used to examine the homogeneity of variances. One-way analysis of variance (ANOVA) was used to analyze the experimental data under different culture and nutrient conditions. Two-factor ANOVA was used to conduct the spore production test. The differences in lowercase letters indicated significant differences in the same medium at different time points ($P < 0.05$), and the differences in uppercase letters indicated substantial differences in different media at the same time points ($P < 0.05$).

## RESULTS

### Phylogenetic analyses

The sequences of the *tef1*, *rpb2*, *tub2*, and *cmdA* loci of seven representative isolates of the FOSC collected from Tuoyaoling Town, Liuhe County in Jilin Province were obtained, besides the sequences of 91 *Fusarium* isolates, including *Fusarium* sp. (CBS 130323), *F. cugenangense* (CBS 130308), *F. foetens* (CBS 120665), and *F. odoratissimum* (CBS 130310), downloaded from GenBank (Table 3). Furthermore, 392 sequences were compared with 2679 nucleotide sites, including 2,304 fixed sites, 194 informational sites, and 181 non-informational sites. In addition, 396 trees were generated using the maximum reduction method, which was similar in topology without significant differences. One of the minimalist trees (TL = 2,407, CI = 0.6129, RI = 0.8398, RC = 0.7602, HI = 0.3871) is shown in Fig. 1.

The seven isolates LH1297, LH1794, LH912, LH532, LH156101, LH1904, and LH1294 from this study formed a branch with CBS 130323 *Fusarium* sp., CBS 680.89 *Fusarium* sp., and CBS 128.81 *Fusarium* sp. with 52% support (Fig. 1).

### Morphology of *F. tobaccum* sp. nov.

Colonies were formed on PDA with an average radial growth rate of 2.2–3.4 mm/day at 25℃. The colony surface was pale wine gray to wine gray and floccose with abundant aerial mycelium. The colony margins were irregular, lobate, serrate, or filiform. The odor was absent. The wine-red center of the colonies with peripheral color gradually decreased, producing diffusible pigment with time. On SNA, mycelia were hyaline, and aerial mycelia were sparse with abundant sporulation and abundant chlamydospores on the medium surface. On CLA, abundant chlamydospores were present on the medium surface, and aerial mycelia were sparse with abundant bright orange sporodochia formed on the carnation leaves. The conidiophores on the aerial mycelia were 35–55 µm tall, unbranched or sparingly branched, bearing terminal or intercalarily phialides, and often reduced to single phialides. The aerial phialides were mono- and 2–4 polyphialidic, subulate to subcylindrical, smooth- and thin-walled, and measured 6.0–15.0 µm × 3.0–4.0 µm, with periclinal thickening inconspicuous or absent. Abundant conidia were formed on PDA and SNA. Aerial conidia formed small false heads on the tips of the

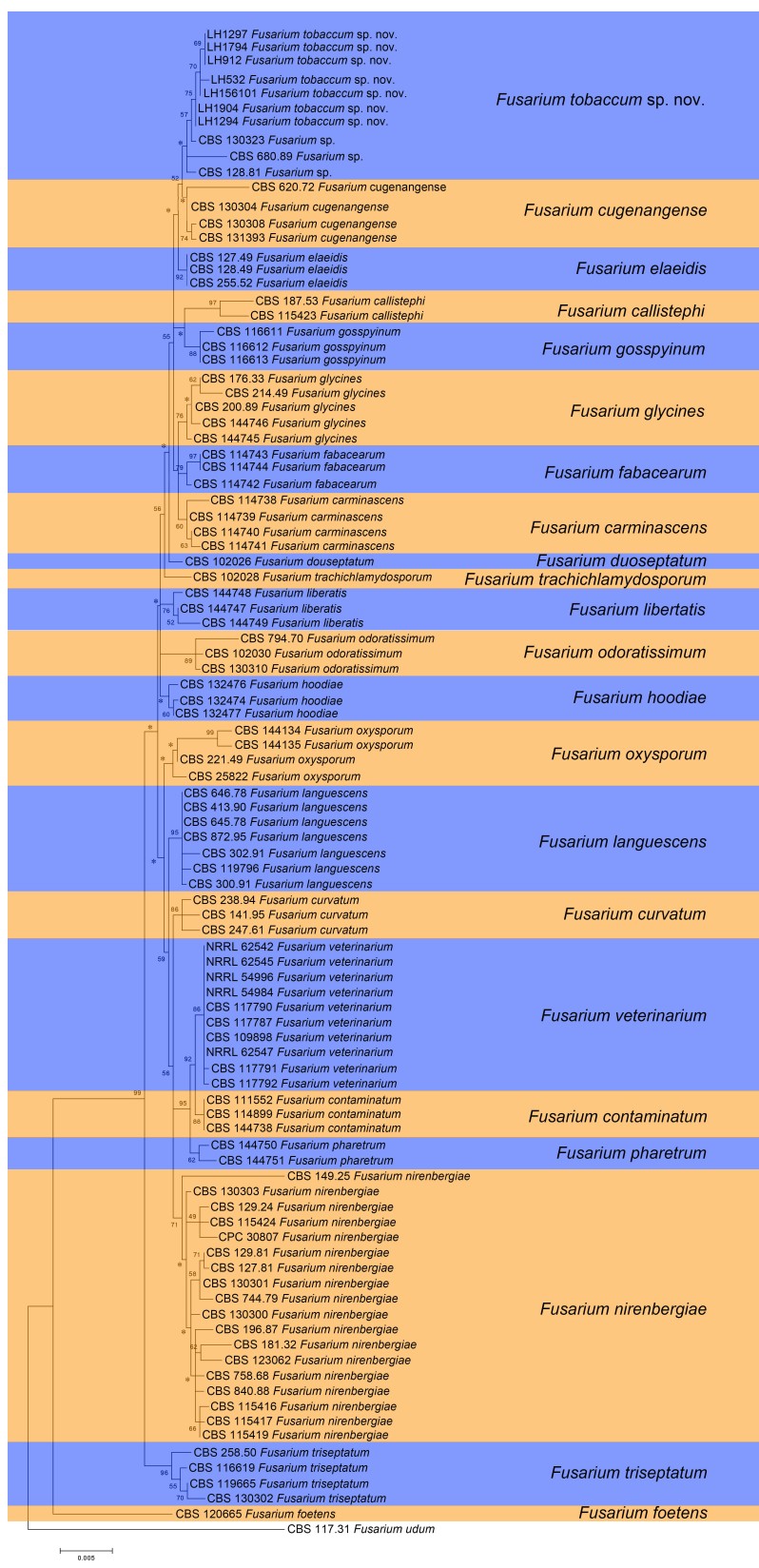

**FIG 1** Maximum parsimony tree generated from the sequence analysis of the concatenated *tef1*, *rpb2*, *tub2*, and *cmdA* data sets. *Fusarium udum* CBS 117.31 was used as an outgroup. Bootstrap support values for maximum parsimony greater than or equal to 50% are shown at the nodes (* indicates value less than 50%).

phialides, and were hyaline, oval to ellipsoid, and 0–1 septate. On SNA, the following were observed: 0-septate conidia: 4.8–6.3 µm × 2.7–3.2 µm (av. 5.5 µm × 3.0 µm); 1-septate conidia: 9.1–10.4 µm × 3.1–3.8 µm (av. 9.9 µm × 3.6 µm). The sporodochia were bright orange and formed sparsely on carnation leaves. The conidiophores in sporodochia were verticillately branched and densely packed, consisted of a short, smooth- and thin-walled stipe, measured 4.0–9.0 µm × 2.0–4.0 µm, bearing apical whorls of two to three monophialides. The sporodochial phialides were subulate to subcylindrical, measured 5–15 µm × 3–4 µm, and were smooth- and thin-walled. Macroconidia were falcate, curved dorsoventrally with almost parallel sides tapering slightly toward both ends (with a blunt to papillate, curved apical cell, and a blunt to foot-like basal cell), falcate, and two to five septate: two-septate conidia: 15.2–19.3 µm × 3.4–3.9 µm (av. 17.8 µm × 3.8 µm); three-septate conidia: 18.9–24.1 µm × 3.4–4.1 µm (av. 22.5 µm × 3.8 µm), four-septate conidia: 29.7–36.4 µm × 3.8–4.5 µm (av. 32.7 µm × 4.1 µm), and five-septate conidia: 31.6–39.7 µm × 4.2–4.7 µm (av. 36.9 µm × 4.5 µm). On CLA, the chlamydospores were smooth-walled, formed terminally or intermediately in a single or tandem chain, and measured 7.6–10.2 µm × 7.7–9.5 µm (av. 9.4 µm × 8.9 µm) (Fig. 2). The pigment area was larger and darker on the back side of the medium compared with that of *F. cugenangense* LH41 (Table 4).

Moreover, a large number of chlamydospores could be produced by *Fusarium tobaccum* sp. nov. in SNA and CLA media for 1–2 weeks, whereas *F. cugenangense* LH41 only produced a small number of chlamydospores in SNA medium for more than 4 weeks. *F. cugenangense* LH41 had a faster mycelial growth rate in PDA medium and formed colonies on PDA with an average radial growth rate of 3.2–3.9 mm/day at 25℃ (Fig. 3).

## Pathogenicity analysis

Disease symptoms were observed on all tobacco plants inoculated with seven isolates on the roots, but no disease symptoms were observed in the control group inoculated without fungus (Fig. 4). The lower parts of the aboveground leaves were yellow and began to wilt 5 days after inoculation (DAI), the upper leaves began to yellow 7 DAI, the lower leaves withered and dried up, and the whole plants wilted 14 DAI. The pathogenicity differences among seven isolates are listed in Table 1. The morphology and DNA sequences of the re-isolated fungi from the inoculated roots were identical to those of the original inoculated isolates.

## Effects of temperature, pH, and light conditions on the mycelial growth of *F. tobaccum* sp. nov.

The mycelia of *F. tobaccum* (*Fto*) sp. nov. LH156101 cultured on PDA for 7 days grew at temperatures ranging from 5℃ to 35℃. The mycelia of *Fto* sp. nov. grew slowly at temperatures from 35℃ to 40℃ ($P < 0.05$), but exhibited the fastest growth at 25℃, with a colony diameter of 70.33 mm. The colony diameter was altered at other temperatures. According to the fitting value of the GaussAmp fit of the relative frequency, $y = y_0 + Ae^{-\frac{(x-x_c)^2}{2w^2}}$, the optimum growth temperature of *Fto* sp. nov. was 23.6℃ (Fig. 5A and C; Table 5).

The mycelia of *Fto* sp. nov. grew at pH values ranging from 6.0 to 11.0, and the colony diameter was the largest (73.67 mm) at pH 7.0. Significant differences in growth diameter were observed at other pH values ($P < 0.05$). The GaussAmp fit of the relative frequency ($y = y_0 + Ae^{-\frac{(x-x_c)^2}{2w^2}}$) indicated that the optimum growth pH value was 7.1; the mycelial growth of *Fto* sp. nov. was slow at pH values ranging from 4.0 to 5.0, and the colony diameter was less than 10 mm after 7 days on PDA (Fig. 5B and D; Table 5).

The effect of light on the mycelial growth of *Fto* sp. nov. was not pronounced. Under full light, darkness, and a 12-h light/12-h dark photoperiod, the mycelia of *Fto* sp. nov. LH156101 grew well on PDA for 7 days at 25℃, with colony diameters of 68.50–72.17 mm. Under full light and a 12-h light/12-h dark photoperiod, the mycelial growth

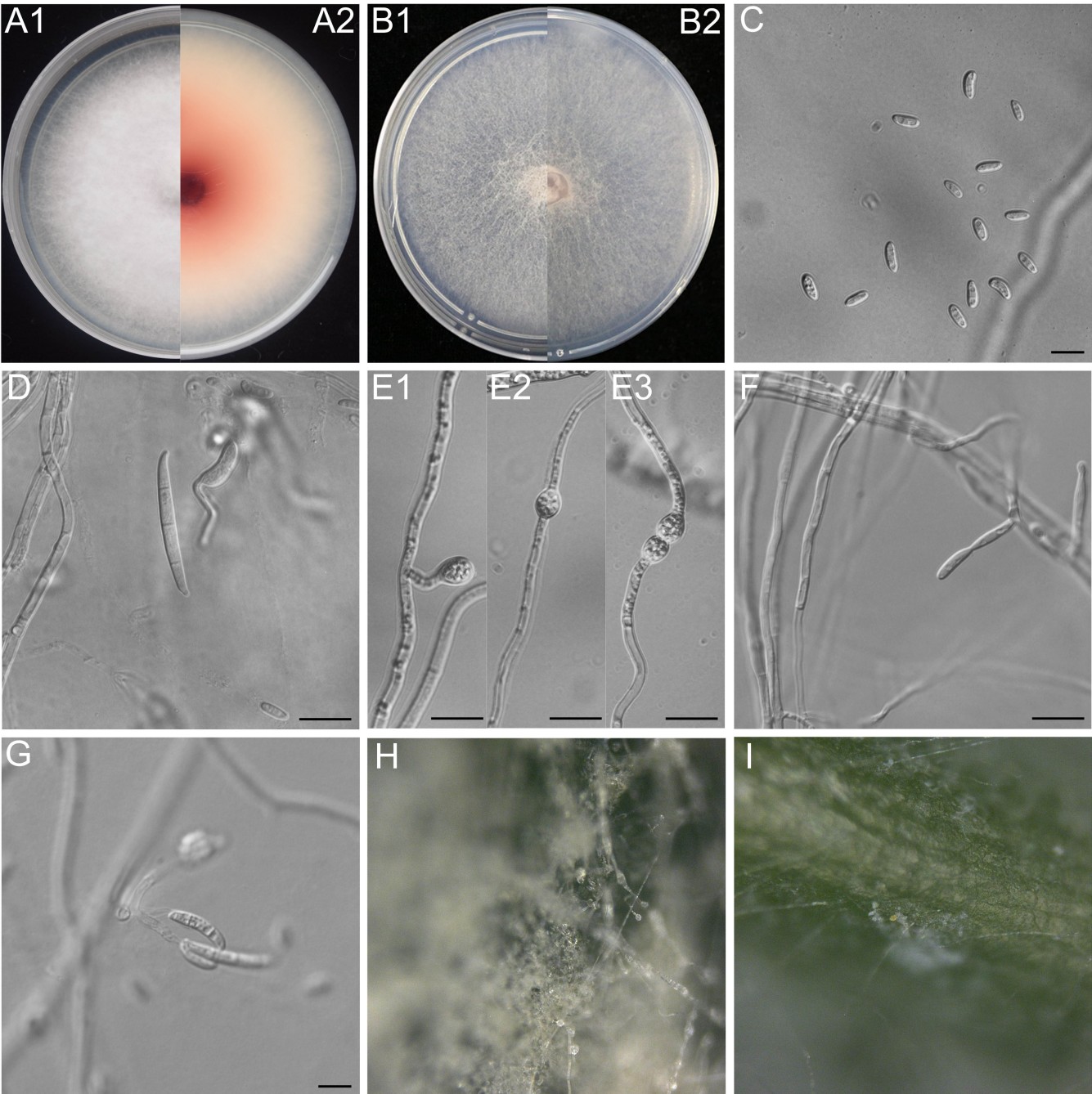

**FIG 2** *Fusarium tobaccum* sp. nov. LH156101. (A) The surface of colonies on PDA after 7 days at 25℃ under continuous white light. (A1) Colonies on PDA. (A2) Back of colonies on PDA. (B) Surface of colonies on SNA after 7 days at 25℃ under continuous white light. (B1) Colonies on SNA. (B2) Back of colonies on SNA. (C) Aerial conidia (microconidia). (D) Sporodochial conidia (macroconidia). (E) Chlamydospores. (F) Conidiophores and phialides on aerial mycelium. (G) False heads carried on phialides on aerial mycelium. (H) Conidiophores on the surface of carnation leaves. (I) Sporodochia on carnation leaves. Scale bars = 10 µm.

of *Fto* sp. nov. was slightly faster than that under darkness ($P < 0.05$) (Fig. 6A and B; Table 5).

## Effects of carbon source, nitrogen source, and medium on the mycelial growth of *F. tobaccum* sp. nov.

The mycelial growth of *F. tobaccum* sp. nov. LH156101 was superior on eight different carbon source media than on media without any carbon source. The growth diameter

TABLE 4  Morphological comparison between *Fusarium tobaccum* sp. nov. and *F. cugenangense*

| Spore | | *Fusarium tobaccum* sp. nov. | *F. cugenangense* |
|---|---|---|---|
| Microconidia | Septate | 0–1 | 0–3 |
| | Size (µm) | 4.8–6.3 × 2.7–3.2 µm | 7.1–9.3 × 2.1–3.2 µm |
| | | (av. 5.5 × 3.0 µm; 0-septate) | (av. 8.5 ×2.8 µm; 0-septate) |
| | | 9.1–10.4 × 3.1–3.8 µm | 13.2–16.8 × 2.3–3.3 µm |
| | | (av. 9.9 × 3.6 µm; 1-septate) | (av. 15.7 × 2.9 µm; 1-septate) |
| | | | 17.1–23.4 × 3.1–4.5 µm |
| | | | (av. 21.4 × 3.7 µm; 2-septate) |
| | | | 21.5–27.6 × 3.2–4.1 µm |
| | | | (av. 24.3 × 3.6 µm; 3-septate) |
| | Shape | Ellipsoidal to falcate | Ellipsoidal to falcate |
| | Substrate/media | PDA/SNA/CLA | PDA/SNA/CLA |
| Macroconidia | Septate | 2–5 | 3–6 |
| | Size (µm) | 15.2–19.3 × 3.4–3.9 µm | 29.8–37.8 × 3.7–4.3 µm |
| | | (av. 17.8 × 3.8 µm; 2-septate) | (av. 34.7 × 4.1 µm; 3-septate) |
| | | 18.9–24.1 × 3.4–4.1 µm | 33.0–40.1 × 3.9–4.6 µm |
| | | (av. 22.5 × 3.8 µm; 3-septate) | (av. 38.5 × 4.2 µm; 4-septate) |
| | | 29.7–36.4 × 3.8–4.5 µm | 35.8–44.6 × 4.1–5.1 µm |
| | | (av. 32.7 × 4.1 µm; 4-septate) | (av. 40.4 × 4.7 µm; 5-septate) |
| | | 31.6–39.7 × 4.2–4.7 µm | 43.4–49.1 × 4.7–5.2 µm |
| | | (av. 36.9 × 4.5 µm; 5-septate) | (av. 45.6 × 5.1 µm; 6-septate) |
| | Shape | Falcate | Falcate |
| | Substrate/media | CLA | CLA |
| Chlamydospores | Shape | Globose to subglobose | Globose to subglobose |
| | Size (µm) | 7.6–10.2 × 7.7–9.5 µm | 10.1–14.2 × 11.3–14.1 µm |
| | | (av. 9.4 × 8.9 µm) | (av. 11.8 × 12.7 µm) |
| | Numbers | Abundance | Few |
| | Time | 1–2 week | 4 weeks |
| | Substrate/media | SNA/CLA | SNA |

of the colony on the medium with sucrose was the largest, with a colony diameter of 59.50 mm. A significant difference was observed between the growth diameter on the medium with sucrose and that on media with other carbon sources ($P < 0.05$), followed by maltose, starch, and trehalose. When the carbon source was maltose and starch, the colony diameter of *F. tobaccum* sp. nov. LH156101 was 57.17 and 57.50 mm, respectively; no significant difference in the colony diameter was observed when the carbon source was maltose and starch ($P < 0.05$) (Fig. 7A and D; Table 5).

The mycelia of *F. tobaccum* sp. nov. LH156101 exhibited the fastest growth on the medium with $NaNO_3$ as the nitrogen source, with a colony diameter of 58.67 mm; a significant difference was observed in the colony diameter on the medium with $NaNO_3$ and that on media with other nitrogen sources ($P < 0.05$). The second highest growth rate was observed on $KNO_3$, and the colony diameter was 44.83 mm. *F. tobaccum* sp. nov. LH156101 grew poorly on the media with the other six compounds as the nitrogen source, as the colony diameter was less than 44.83 mm after culture at 25°C for 7 days. The differences in growth between these six nitrogen sources and growth on the medium without any nitrogen source were small. The lack of a nitrogen source had a significant effect on the growth of the aerial mycelia than the lack of a carbon source (Fig. 7B and E; Table 5).

The mycelium of *F. tobaccum* sp. nov. LH156101 exhibited the fastest growth on the PDA medium with a colony diameter of 71.33 mm. A significant difference was observed between the growth diameter of colonies on PDA and that on other media ($P < 0.05$), including Czapek medium (Fig. 7C and F; Table 5). Media had a significant effect on the production of aerial mycelia than the carbon source.

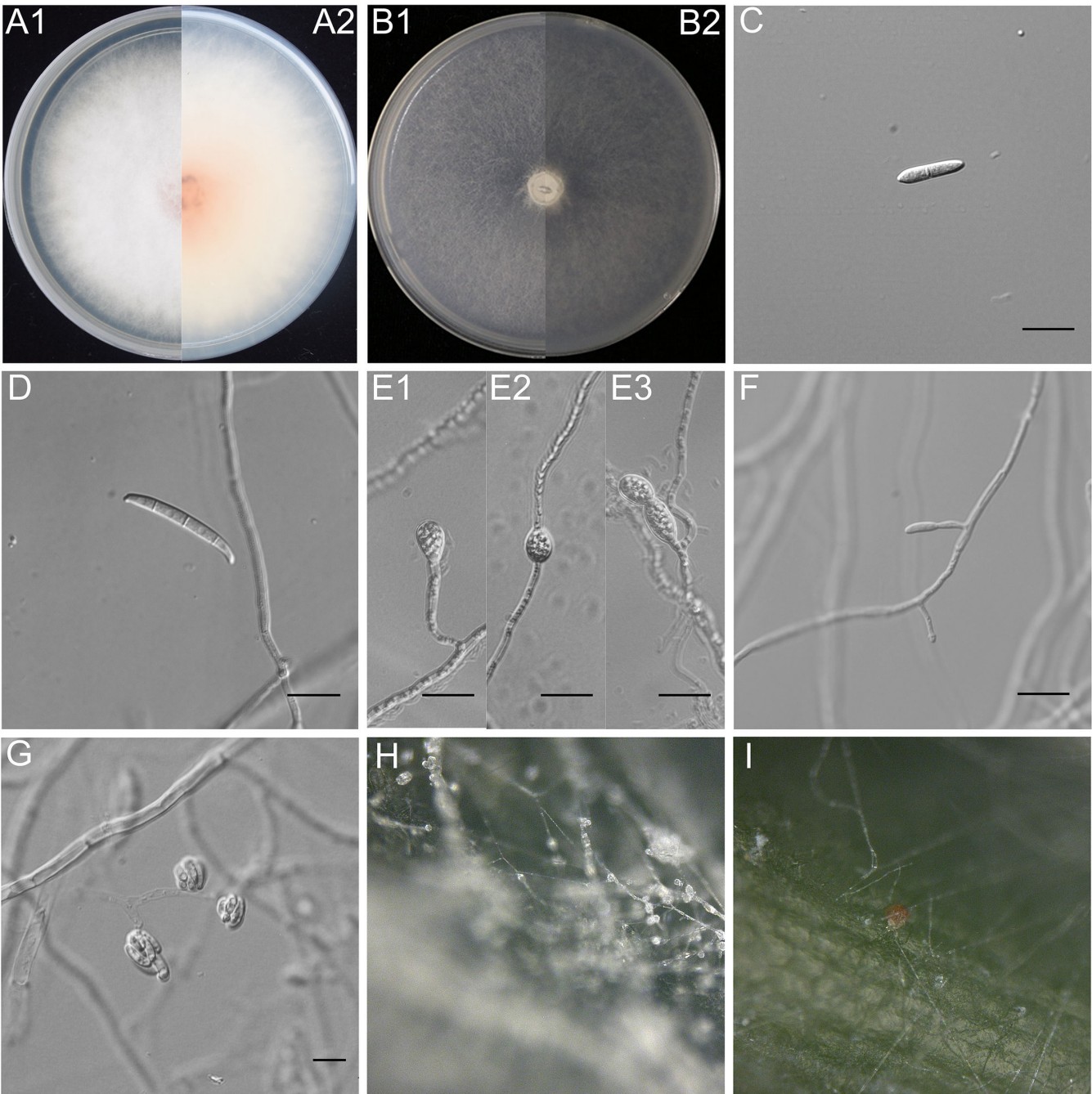

**FIG 3** *Fusarium cugenangense* LH41. (A) The surface of colonies on PDA after 7 days at 25°C under continuous white light. (A1) Colonies on PDA. (A2) Back of colonies on PDA. (B) Surface of colonies on SNA after 7 days at 25°C under continuous white light. (B1) Colonies on SNA. (B2) Back of colonies on SNA. (C) Aerial conidia (microconidia). (D) Sporodochial conidia (macroconidia). (E) Chlamydospores. (F) Conidiophores and phialides on aerial mycelium. (G) False heads carried on phialides on aerial mycelium. (H) Conidiophores on the surface of carnation leaves. (I) Sporodochia on carnation leaves. Scale bars = 10 µm.

## Sporulation of *F. tobaccum* sp. nov. in PD, SN, and CMC liquid media

*F. tobaccum* sp. nov. LH156101 was cultured at 25°C in PD, SN, and CMC. The conidia were produced after 36 h, and sporulation increased with the extension of the culture time. The spore yield of the isolates cultured in PD, CMC, and SN for 7 days at 25°C was $13.39 \times 10^6$ spores/mL, $11.31 \times 10^6$ spores/mL, and $8.12 \times 10^6$ spores/mL, respectively. Among the three media, PD was optimal for the sporulation of *F. tobaccum* sp. nov. LH156101, followed by CMC and SN (Fig. 8A; Table 6).

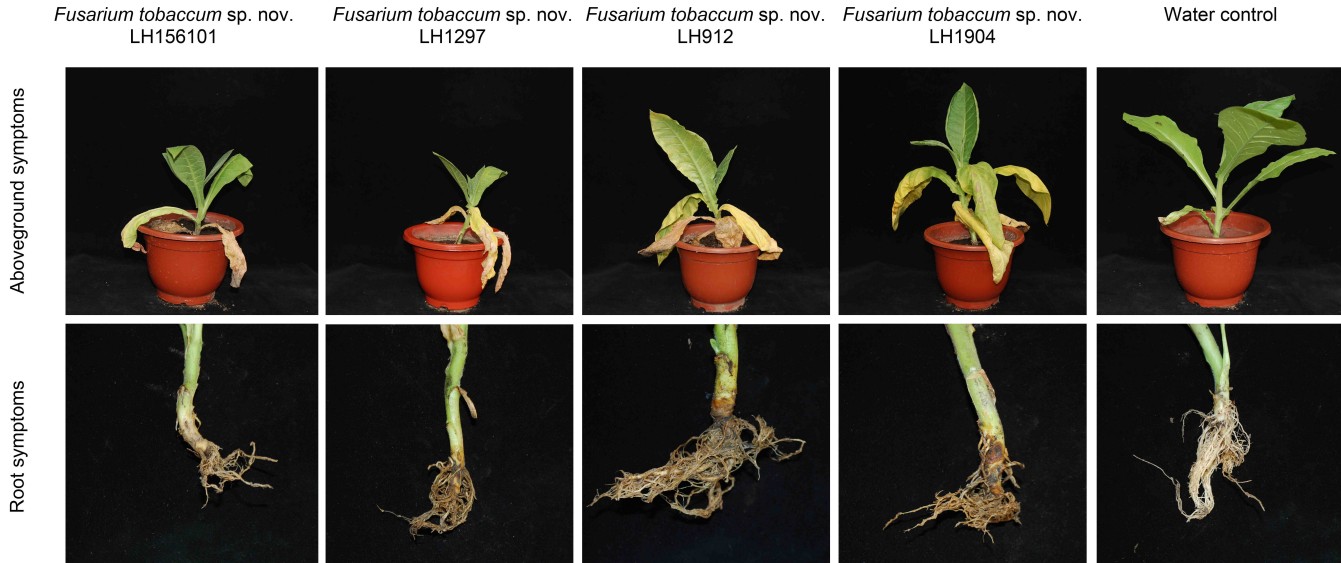

**FIG 4**  Pathogenicity of representative isolates of *F. tobaccum* sp. nov. inoculated with barley grain–fungus culture on tobacco roots *in vivo*.

*F. tobaccum* sp. nov. LH156101 began to produce macroconidia cultured in PD, SN, and CMC media after 36 h at 25℃. Although *F. tobaccum* sp. nov. LH156101 could produce macroconidia in PD, SN, and CMC, the proportion of macroconidia produced in the three media was low. Although the proportion of macroconidia produced was higher in SN and CMC than in PD, the proportion of macroconidia produced in PD, SN, and CMC was only 0.48%–2.16% in 7 days at 25℃ compared with that in 36 h at 25℃ (0.68%, 1.48%, and 0.96%, respectively). The proportion of macroconidia produced in PD, CMC, and SN did not significantly increase with the extension of the culture time. In general, the PD culture medium was not conducive to the production of macroconidia, and macroconidia production was the lowest in the PD medium among the three media. The proportion of macroconidia was less than 0.68%. In summary, the proportion and total number of macroconidia of *F. tobaccum* sp. nov. LH156101 in the three media were low (Fig. 8B; Table 6).

## DISCUSSION

In this study, the seven isolates collected from diseased tobacco plants in Jilin Province, China, were identified as members of a new species, *F. tobaccum* sp. nov., *via* analysis of morphological characteristics and phylogenetic analyses based on the *tef1*, *tub2*, *cmdA*, and *rpb2* genes. Moreover, the biological characteristics, including the optimal pH, temperature, carbon source, and nitrogen source, were also determined.

*F. oxysporum* is the most significant plant pathogenic soil-borne asexual fungus, ranking fifth among the 10 most economically significant fungal pathogens (56, 57). *F. oxysporum* is typically considered a species complex with at least 26 distinct species and clades (38, 58, 59). Recently, Lombard (42) identified a new epitype of *F. oxysporum* and named 15 cryptic species in the FOSC based on multi-locus phylogenetic analyses and subtle morphological differences. In our study, seven isolates of FOSC obtained from diseased tobacco roots were used to construct phylogenetic trees based on four gene sequences (*tef1*, *tub2*, *cmdA*, and *rpb2*). The seven isolates of FOSC were grouped into a clade with *Fusarium* sp. CBS 130323, CBS 128.81, and CBS 680.89 in the phylogenies constructed using four gene sequences. According to Lombard (42), the three isolates of *Fusarium* sp., CBS 130323, CBS 680.89, and CBS 128.81, were not named because the data on their morphology and biological characteristics were lacking. In this study, the seven isolates pathogenic to tobacco grouped in a clade with *Fusarium* sp. CBS 130323, CBS 128.81, and CBS 680.89 were identified as a new species based on the phylogenetic

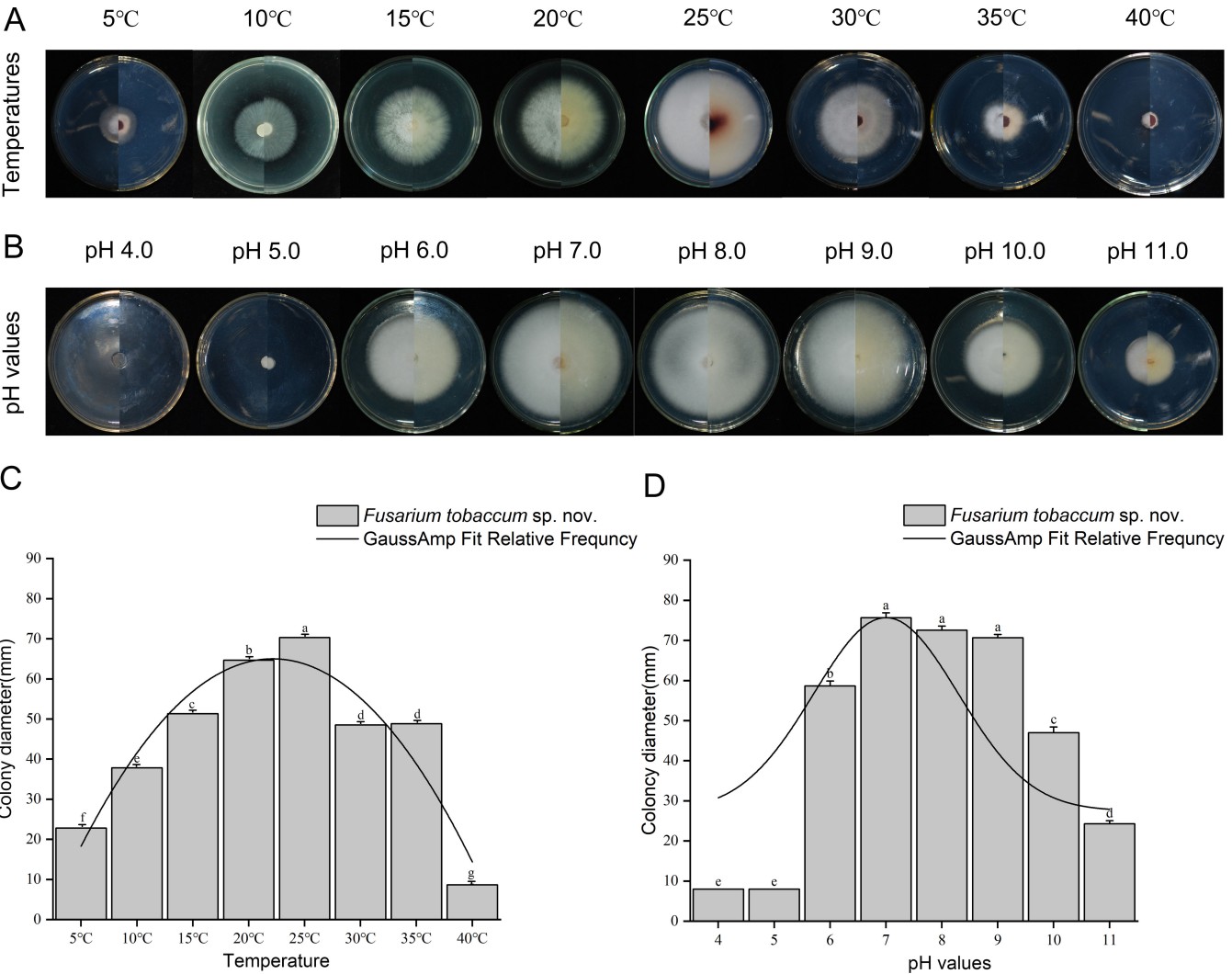

**FIG 5** Colony morphology and diameters of *F. tobaccum* sp. nov. LH156101 on PDA after 7 days of culture at different temperatures (A) and pH (B). Bars with different lowercase letters (above column diagram) indicate significant differences according to Duncan's multiple range tests at the $P < 0.05$ level (C and D).

analyses of four genes and their pathogenicity to tobacco, which differed from that of other species within the FOSC that were identified as causal agents of FWT.

The colony morphology, pigment production, and spore characteristics are important features for distinguishing among members of the FOSC. According to the phylogenetic tree based on the *tef1*, *tub2*, *cmdA*, and *rpb2* gene sequences, *F. tobaccum* sp. nov. is closely related to *F. cugenangense*, *F. callistephi,* and *F. elaeidis*. However, the four species differ in morphology. In comparison to *F. callistephi* and *F. elaeidis* (42), the aerial mycelia of both *F. tobaccum* sp. nov. and *F. cugenangense* were abundant on PDA, *F. tobaccum* sp. nov. produced more pigment on PDA medium and eventually diffused across the entire Petri dish (Fig. 2). It is worth noting that *F. tobaccum* sp. nov. was capable of producing abundant chlamydospores on SNA and CLA media for 1–2 weeks, either singly or in tandem chains. However, *F. cugenangense* only produced a small number of chlamydospores after 4 weeks of culture on SNA medium, and chlamydospores were not observed in *F. callistephi* and *F. elaeidis*. In addition, *F. tobaccum* sp. nov. produced microconidia with 0–1 septa and macroconidia with 2–5 septa compared with *F. cugenangense*, *F. callistephi,* and *F. elaeidis* (42) which had microconidia with 0–3/0-1/0-1 septa and macroconidia with 3–6/3-5/1-5 septa. These findings, combined with the molecular results, indicated that the seven isolates derived from tobacco roots

**TABLE 5** Colony diameters of *F. tobaccum* sp. nov. LH156101 after 7 days under different conditions[a]

| Cultural condition | Treatments | | | | | | | |
|---|---|---|---|---|---|---|---|---|
| Temperature | 5°C | 10°C | 15°C | 20°C | 25°C | 30°C | 35°C | 40°C |
| Colony diameter (mm, mean ± SE[b]) | 22.83 ± 0.69f | 37.83 ± 1.34e | 51.33 ± 0.94c | 64.67 ± 0.47b | 70.33 ± 1.37a | 48.50 ± 1.89d | 48.83 ± 1.80d | 8.67 ± 0.94g |
| pH value | 4.0 | 5.0 | 6.0 | 7.0 | 8.0 | 9.0 | 10.0 | 11.0 |
| Colony diameter (mm, mean ± SE) | 8.00 ± 0.00e | 8.00 ± 0.00e | 57.17 ± 1.34b | 73.67 ± 0.86a | 71.07 ± 0.74a | 69.83 ± 0.94a | 47.00 ± 1.41c | 25.17 ± 0.58d |
| Light conditions | Full exposure | Total dark | Alteration of light and darkness | | | | | |
| Colony diameter (mm, mean ± SE) | 72.17 ± 0.69a | 68.50 ± 1.50b | 72.00 ± 0.58a | | | | | |
| Carbon source | CK (carbon free) | Sucrose | Lactose | Maltose | Trehalose | Starch | Mannitol | Glycerin |
| Colony diameter (mm, mean ± SE) | 47.17 ± 2.11e | 59.50 ± 2.36a | 51.67 ± 2.29c | 57.17 ± 1.34b | 50.00 ± 1.91d | 57.50 ± 0.76b | 53.33 ± 1.11c | 50.50 ± 0.96d |
| Nitrogen source | CK (nitrogen free) | NaNO₃ | NH₄NO₃ | KNO₃ | Peptone | Beef extract | Cystine | Aspartic acid |
| Colony diameter (mm, mean ± SE) | 36.83 ± 0.69c | 58.67 ± 0.75a | 34.33 ± 0.94d | 44.83 ± 1.07b | 36.83 ± 0.69c | 36.67 ± 1.11c | 34.67 ± 0.94d | 36.67 ± 0.94c |
| Media | AEA | Czapek | OA | PCA | PDA | PSA | SNA | V8 |
| Colony diameter (mm, mean ± SE) | 49.00 ± 1.15e | 50.50 ± 0.50d | 55.33 ± 0.94c | 55.67 ± 0.75c | 71.33 ± 0.47a | 56.67 ± 1.11c | 55.83 ± 0.90c | 59.17 ± 1.21b |

[a]Means within the column followed by the same letter(s) are not significantly different from those assessed by Duncan's multiple range tests (*P* ≤ 0.05).
[b]The data presented here are the mean value ± SE (standard error) of three replicates in each treatment.

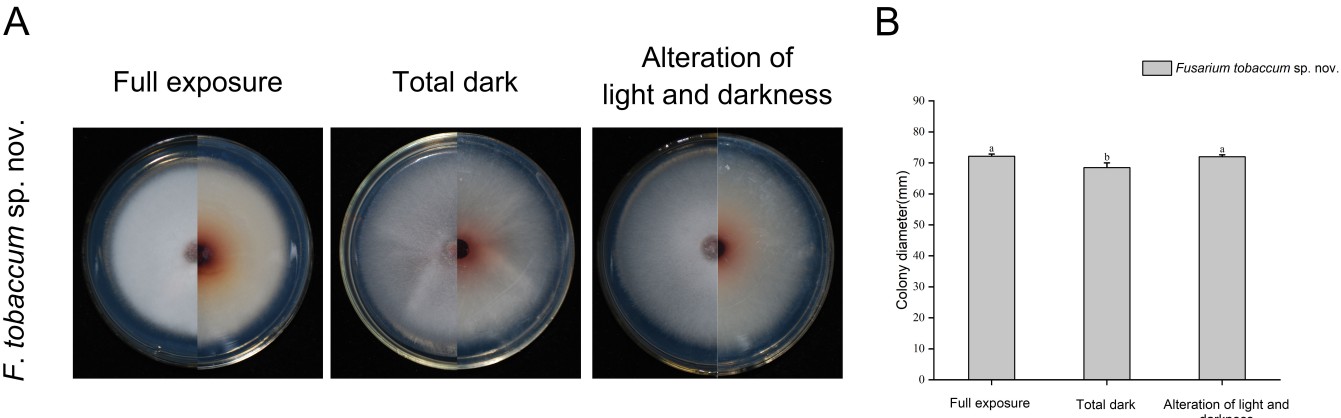

**FIG 6** Colony morphology (A) and colony diameters (B) of *F. tobaccum* sp. nov. LH156101 cultured under different light conditions for 7 days. Bars with different lowercase letters (above column diagram) indicate significant differences according to Duncan's multiple range tests at the *P* < 0.05 level (B).

were identified as *F. tobaccum* sp. nov. Zhao Xie & Jie Gao. This was the first study of *F. tobaccum* causing the root rot of tobacco.

**FIG 7** Colony morphology and diameters of *F. tobaccum* sp. nov. LH156101 after 7 days of culture at different temperatures on different carbon source media (A), nitrogen source media (B), and media (C). Bars with different lowercase letters (above column diagram) indicate significant differences according to Duncan's multiple range tests at the *P* < 0.05 level (D–F).

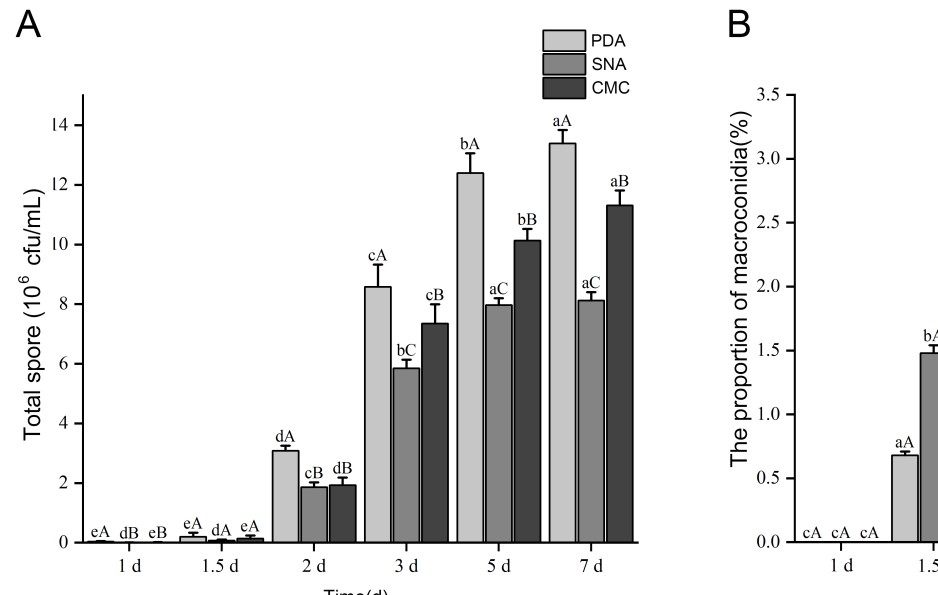
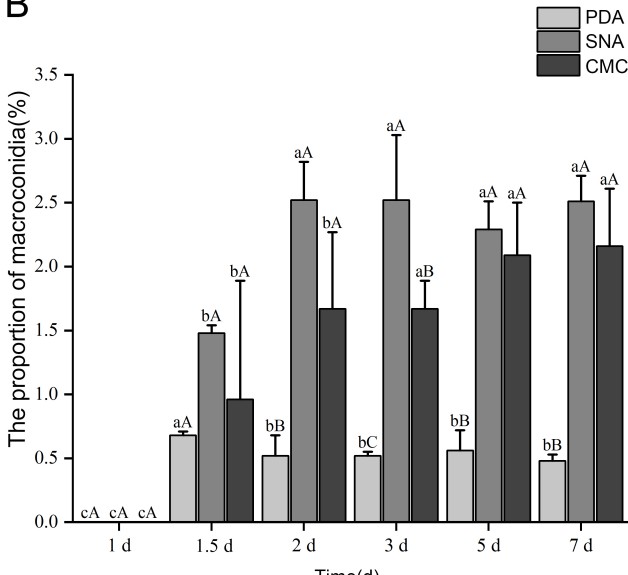

**FIG 8** Effects of media and incubation time on the conidiation (A) and proportion of macroconidia (B) of *F. tobaccum* sp. nov. LH156101. Bars with different uppercase letters (among media) or lowercase letters (among different time points) indicate significant differences according to Duncan's multiple range tests at the *P* < 0.05 level.

The investigation of biological characteristics is essential for developing methods to control tobacco root rot caused by *F. tobaccum* sp. nov. Few studies have been conducted on the biological characteristics of *F. oxysporum*. A study investigated the effects of environmental and nutrient factors, such as temperature, pH, carbon source, and nitrogen source, on the mycelial growth of *F. tobaccum* sp. nov. The results showed that the optimum temperature for the mycelial growth of *F. tobaccum* sp. nov. was 23.6°C, and the optimum pH for mycelial growth was 7.1. This slightly differed from the biological characteristics of *F. oxysporum* isolated from *Atractylodes macrocephala* (60) and *F. oxysporum* f. sp. *vanilla* (61), which had optimal mycelial growth temperatures of 28°C and 25–30°C and pH of 6.0 and 6.5, respectively. These differences might be related to the species of FOSC and hosts. In addition, the mycelia of *F. tobaccum* sp. nov. exhibited the fastest growth when sucrose was used as the carbon source. Nitrogen also had a significant effect on the growth rate of mycelia; NaNO$_3$ resulted in the fastest growth, followed by KNO$_3$. Light conditions had little effect on the mycelial growth of *F. tobaccum* sp. nov., which was contrary to *F. oxysporum* f. sp. *vanillae* (61) growing fastest under alternating light and dark conditions.

*F. tobaccum* sp. nov. produced a large number of spores when it was cultured in PD for 3–7 days at 25°C. However, the proportion of macroconidia was low in PD, SN, and CMC media, and the lowest proportion was observed in the PD medium. Only the effects of

**TABLE 6** Effects of media and incubation time on the conidiation and proportion of macroconidia of *F. tobaccum* sp. nov. LH156101[a]

| Media | Conidium | Time (d) | | | | | |
|---|---|---|---|---|---|---|---|
| | | 1 | 1.5 | 2 | 3 | 5 | 7 |
| PD | Conidiation (mean ± SE[b] × 10$^6$ spores/mL) | 0.04 ± 0.01eA | 0.20 ± 0.13eA | 3.08 ± 0.17cA | 8.58 ± 0.75cA | 12.40 ± 0.66bA | 13.39 ± 0.45aA |
| SN | Conidiation (mean ± SE × 10$^6$ spores/mL) | 0.00 ± 0.01dB | 0.07 ± 0.03dA | 1.86 ± 0.16cB | 5.85 ± 0.29bC | 7.97 ± 0.23aC | 8.12 ± 0.28aC |
| CMC | Conidiation (mean ± SE × 10$^6$ spores/mL) | 0.01 ± 0.01eB | 0.14 ± 0.10eA | 1.93 ± 0.25dB | 7.35 ± 0.64cB | 10.13 ± 0.39bB | 11.31 ± 0.5aB |
| PD | Proportion of macroconidia (%) | 0.00 ± 0.00cA | 0.68 ± 0.03aA | 0.52 ± 0.16bB | 0.52 ± 0.03bC | 0.56 ± 0.16bB | 0.48 ± 0.05bB |
| SN | Proportion of macroconidia (%) | 0.00 ± 0.00 cA | 1.48 ± 0.06bA | 2.52 ± 0.30aA | 2.52 ± 0.51aA | 2.29 ± 0.22aA | 2.51 ± 0.20aA |
| CMC | Proportion of macroconidia (%) | 0.00 ± 0.00 cA | 0.96 ± 0.93bA | 1.67 ± 0.60bA | 1.67 ± 0.22aB | 2.09 ± 0.41aA | 2.16 ± 0.45aA |

[a]Notes: Means within the column followed by the same letter(s) are not significantly different from those assessed by Duncan's multiple range tests (*P* ≤ 0.05).
[b]The data presented here are the mean value ± SE (standard error) of three replicates in each treatment.

three liquid cultures and culture time on the production of conidia and the proportion of macroconidia of *F. tobaccum* sp. nov. were investigated. Additional studies are needed to clarify the effects of other nutrients (carbon and nitrogen sources, different media) and environmental conditions (pH and temperature) on conidia, especially the macroconidia production, of *F. tobaccum* sp. nov.

In summary, *F. tobaccum* sp. nov. is a novel species in the FOSC that causes FWT in China. The effects of temperature, pH, carbon source, nitrogen source, media, and light conditions on the mycelial growth of *F. tobaccum* sp. nov. were determined. *F. tobaccum* sp. nov. could produce large conidia in PD, SNA, and CMC, but the number of macroconidia in the three media was low. These findings might help in further exploring the pathogenic mechanism of the new species and the development of strategies for controlling FWT.

## Conclusions

In this study, seven Fusarium isolates isolated from diseased samples in Liuhe County, Jilin Province, China, were identified as a new species, *Fusarium tobaccum* sp. nov. Zhao Xie & Jie Gao based on multi-locus sequence analysis of *tef1-rpb2-tub2-cmdA*, combined with morphological characteristics. *F. tobaccum* sp. nov. exhibits abundant aerial mycelia and pigment production on PDA, microconidia with 0–1 septa, and macroconidia with 2–5 septa on CLA, and produces abundant chlamydospores on SNA and CLA. According to the pathogenicity tests, these seven Fusarium isolates were all pathogenic to tobacco roots and could be responsible for FWT. In addition, the effects of temperature, pH, carbon source, nitrogen source, and light on the mycelial growth were also revealed. The findings in this work confirm *F. tobaccum* sp. nov. as a novel species capable of causing FWT and will be helpful for a better understanding of the morphological and biological characteristics of *Fusarium*.

### ACKNOWLEDGMENTS

This study was supported by the major special project for green control of tobacco pests in Jilin Province (2022220000200135) and the overseas expertise Introduction project for the discipline Innovation (111 project) (D17014).

### AUTHOR AFFILIATIONS

[1]College of Plant Protection, Jilin Agricultural University, Changchun, China
[2]Institute of tobacco, Yanbian Academy of Agricultural Science, Yanji, China
[3]State-Local Joint Engineering Research Center of Ginseng Breeding and Application, Changchun, China

### AUTHOR ORCIDs

Yanjing Zhang  http://orcid.org/0000-0003-3663-5633
Jie Gao  http://orcid.org/0000-0003-2021-819X

### FUNDING

| Funder | Grant(s) | Author(s) |
| --- | --- | --- |
| Major special project for green control of tobacco pests in Jilin Province | 2022220000200135 | Jie Gao |
| SCUT \| Overseas Expertise Introduction Center for Discipline Innovation of Food Nutrition and Human Health (111 Center) | D17014 | Jie Gao |

## DATA AVAILABILITY

The raw sequence reads of *Fusarium tobaccum* sp. nov. have been submitted to the GenBank database under accession numbers OM162136, OM162140, OM162144, MN610673, OM162148, OM162150, MN610680, OL437357, OL437361, OL437365, OL437366, OL437371, OL437372, OL437369, OL437329, OL437333, OL437337, OL437338, OL437343, OL437344, OL437341, OL437385, OL437389, OL437393, OL437394, OL437399, OL437400, and OL437397.

## ADDITIONAL FILES

The following material is available online.

### Open Peer Review

**PEER REVIEW HISTORY (review-history.pdf).** An accounting of the reviewer comments and feedback.

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
