## [Reviewer comments · Microbiology Spectrum]

Microbiology Spectrum

Identification and biological characteristics of *Fusarium tobaccum* sp. nov., a novel species causing tobacco root rot in Jilin Province, China

Zhao Xie, Chong Gao, Xiaoyan Zhang, Mengzhu Du, Jun Wang, Xue Wang, Baohui Lu, Changqing Chen, Lina Yang, Yanjing Zhang, and Jie Gao

Corresponding Author(s): Jie Gao, Jilin Agricultural University

Review Timeline:

Submission Date:	April 18, 2024
Editorial Decision:	June 22, 2024
Revision Received:	July 14, 2024
Editorial Decision:	August 5, 2024
Revision Received:	August 11, 2024
Accepted:	August 20, 2024

Editor: Jonathan Snow

Reviewer(s): Disclosure of reviewer identity is with reference to reviewer comments included in decision letter(s). The following individuals involved in review of your submission have agreed to reveal their identity: Laith Khalil Tawfeeq Al-Ani (Reviewer #1); Chunwei Wang (Reviewer #2)

Transaction Report:

DOI: <https://doi.org/10.1128/spectrum.00925-24>

Re: Spectrum00925-24 (Identification and biological characteristics of *Fusarium tobaccum* sp. nov., a novel species causing tobacco root rot in Jilin Province, China)

Dear Prof. Jie Gao:

Thank you for transferring your manuscript to Microbiology Spectrum. Two reviewers have read and commented on your manuscript. Both make suggestions for improvements, clarifications and corrections that I encourage you to consider.

Revision Guidelines

Sincerely,
Jonathan Snow
Editor
Microbiology Spectrum

Reviewer #1 (Comments for the Author):

1. Line 51-53:

The aim of study is poorly and need to write in clear way

2. the title of sections in material and methods:
not sound and change it to be suitable as paper

3. Line 56: "Diseased FWT-like samples were collected between 2018 from five towns spanning four counties or cities"
Rewrite

4. Change all term "Strain" to "isolate"

5. What is meant = "macrospores" in line 156 and in conclusion ?

6. intercalarily in line 191:
What is this? correct it

7. *F. cugenangense*:
What is this? Why are mentioned to this species? but you took on one

8. CONCLUSIONS:
Poorly, and need to rewrite

9. Line 192: "polyphialidic,"
This characterize is not associated with *F. oxysporum* morphology

Reviewer #2 (Comments for the Author):

1. *Fusarium wilt of tobacco root rot (FWT) or Fusarium wilt of tobacco (FWT)?* FWT was different between ABSTRACT and IMPORTANCE.

2. *Fusarium tobaccum sp. nov.* could cause tobacco root rot OR *Fusarium wilt of tobacco*? Please uniform it.

3. In line 55, "between 2018" might be error grammar. Please revise it.

4. In line 55, diseased samples were collected between 2018 from five towns spanning four counties or cities in Jilin. However, *Fusarium tobaccum sp. nov.* were only isolated from Tuoyaoling town, Liuhe county? Please confirm it.

5. Longitude and latitude of sampling site should be provided in Table 1.

6. In Line 57, authors give this sentence "A total of 132 pure cultures were obtained and seven strains were further studied (Table 1)". However, seven strain were only presented in Table 1. Please make it clear.

7. In Line 61, Media used in this study were provide in this section. Please give suitable References to support each Media formula. In addition, pH value is important for the culture of the tested strains. Please give pH value for Media.

8. In Line 134, among seven strains, authors have selected strain LH156101 for biological characteristics, why?

9. In Line 329, the sentence of "Effects of pH, temperature, media, carbon source, nitrogen source and light on the mycelial growth of *F. tobaccum sp. nov.* were determined;" is not suitable in CONCLUSIONS. Please provide the optimal conditions for *F. tobaccum sp. nov.* In addition, this sentence was repeated with the sentence in Line 320.

Response to the editors' and the reviewers' comments

Dear editors and reviewers,

We appreciate the valuable comments and suggestions from you and the reviewers to our manuscript entitled “Identification and biological characteristics of *Fusarium tobaccum* sp. nov., a new species causing tobacco root rot in Jilin Province, China” (ID: #AEM01776-23). We have revised the manuscript carefully and provided a point-by-point response to the review’s comments. We have also invited an English native speaker to revise the manuscript carefully. The modifications according to the reviewers’ suggestions are highlighted in yellow. And other modifications mainly in English language are in red in the manuscript.

Finally, we think the modifications have improved the manuscript and thank you very much for your attention to our manuscript. Hopefully we have answered all your questions and please let us know if you have any further concerns.

Sincerely yours,

Jie Gao, Ph. D & Professor

Jilin Agricultural University

jjegao115@126.com

1. Response to comments/suggestion of the reviewer #1s' comments

Comment #1: Was the difference between the new species and other species?

Response: Thank you for your question. According to the phylogenetic tree based on the *tef1*, *tub2*, *cmdA*, and *rpb2* gene sequences, *F. tobaccum* sp. nov. is closely related to *F. cugenangense*. However, the two species differ in morphology. Although the aerial mycelia of both *F. tobaccum* sp. nov. and *F. cugenangense* were abundant on PDA, *F. tobaccum* sp. nov. produced more pigment on PDA medium and eventually occupied the entire Petri dish (Fig. 3), and it was capable of producing chlamydo spores on SNA and CLA media, either singly or in tandem chains. However, *F. cugenangense* only produced a small number of chlamydo spores after 4 weeks of culture on SNA medium. In addition, *F. tobaccum* sp. nov. produced microconidia with 0–2 septa and macroconidia with 3–5 septa compared with *F. cugenangense*, which have microconidia with 0–3 septum and macroconidia with 3–6 septa. These

findings, combined with the molecular results, indicate that the isolates derived from tobacco roots identical to *Fusarium* sp. represent *F. tobaccum* sp. nov. Zhao Xie & Jie Gao. To our knowledge, this is the first report of *F. tobaccum* causing root rot of tobacco. The difference between the *F. tobaccum* sp. nov. and *F. cugenangense* were summarized in Table 4.

Comment #2: Figure 3. "C Aerial conidia (microconidia)", please change as "C. Aerial conidia (microconidia)".

Response: Thanks for your suggestion. We have unified the format of Figure legends. Therefore, we have changed it into "(C) Aerial conidia (microconidia)".

Comment #3: In the title "3.5 Effect of temperatures, pH values, and light conditions on the growth of *F. tobaccum* sp. nov." The "sp." should not be written as italic.

Response: Thanks for your suggestion. It has been revised.

Comment #4: In the paragraph "3.7 Sporulation of *F. tobaccum* sp. nov. in PD, SN, and CMC liquid media". The word "cfu/mL", whether the "cfu" is correct?

Response: Thank you for your suggestion. The "cfu/mL" has been changed as "spores/mL" in the text.

Comment #5: In the research on the biological characteristics of the new species, only the effects of nutritional conditions and environmental factors on mycelial growth were observed, why did not the author observe the effect on sporulation of the new species?

Response: We greatly appreciate your suggestions. In nutritional conditions and environmental factors on mycelial growth experiment, we also tried to obtain the data for the effects of different environmental factor and nutritional conditions on sporulation, but the results were not repeatable. So we did not present the results. However, we compared the effect of three media, PD, SN, and CMC, on the total number of spores produced and the ratio of macrospores of this species in the part "Sporulation of *F. tobaccum* sp. nov. in PD, SN, and CMC liquid media.

Comment #6: What is the basis for the author's selection of *tef1*, *tub2*, *cmdA*, and *rpb2* gene loci to identify new species under FOOSC?

Response: Thank you for your insightful suggestion. In 2019, Lombard et al. used the *tef1*, *tub2*, *cmdA*, and *rpb2* gene to distinguish different species of *Fusarium oxysporum* species complex (FOSC). In our study, we construct the phylogenetic tree based on one of the four genes *tef1*, *tub2*, *cmdA*, and *rpb2* or the multiple *tef1-tub2-cmdA-rpb2* loci, indicating that multiple gene phylogenetic analysis is most effective for distinguishing different species of FOSC. Finally, the *tef1*, *tub2*, *cmdA*, and *rpb2* gene loci were selected to identify new species under FOSC.

Comment #7: In the pathogenicity test, the author directly used the barley-fungus inoculation method, why did not choose other methods?

Response: Thank you for your suggestion. We initially used inoculation methods of root soaking in spore suspension of *Fusarium* spp. and root injury inoculation methods, but their repeatability was not good. Later, we referenced to the method "barley seed-fungus inoculation" from Fang, S. H. *et al.* in reference (50). Our repeated experiments confirmed that the barley seed-fungus inoculation method was the most effective. Therefore, the barley seed-fungus inoculation method was used in this study.

Comment #8: Table3, *F. tobaccum* sp. nov, the full name of *Fusarium* should be written when first appearing in the table.

Response: Thank you for your suggestion. According to the reviewer #2s' comments, the order of the species or isolates in Table 3 was reorganized. So the first species *Fusarium callistephi* was written as full name of *Fusarium*.

Comment #9: Figure 8 and Figure 9. at the P<0.05 level (D, E, F), the "P " should be italic.

Response: Thank you for your suggestion. We have revised them in the manuscript.

Comment #10: Figure 9. In the sentence "Effects of media and incubation time on the conidiation (A) and proportion of macrospores (B) of *F. tobaccum* sp. nov. LH156101. Bars with different uppercase letters (among media) or lowercase letters (among different time points) indicate significant differences according to HSD tests at the P<0.05 level". The "*F. tobaccum*" should be italic.

Response: Thank you for your suggestion, we have changed it in the Figure 9.

Comment #11: References

1. Cole, JS., and Zvenyika, ZA. Stem and root rot of tobacco transplants in Zimbabwe caused by *Rhizoctonia solani* Kuhn and *Fusarium solani* (Mart.) Sacc. Zimbabwe Journal of Agricultural Research. 20:149-152(1982). The " Kuhn" should be " Kühn".

Response: Thank you for your suggestion, we have changed it in the text.

2. Response to comments/suggestion of the reviewer #2s' comments

Comment #1: Table 1 contains GenBank accession numbers for *tef1* sequences from various isolates, yet the authors fail to explicitly mention this in the table or corresponding text. It is crucial to clarify the nature of the provided accession numbers to enhance reader understanding. Additionally, there is an inconsistency where both isolates LH1294 and LH1297 share the same accession number, MN610673. This inconsistency is also observed in Table 3. The authors should address this issue by providing unique accession numbers for each isolate or offering a clear explanation for the shared accession number. Improving the presentation of this information will contribute to the overall clarity and reliability of the manuscript.

Response: We are sorry for these errors. We've supplemented the corresponding descriptions, and changed the accession number OM162144 for the strain LH1294 and the accession number MN610673 for the strain LH1297.

Comment #2: The presentation of Table 3 in the manuscript requires improvement. A notable concern is the lack of citations for information primarily sourced from Lombard et al. Additionally, the fungal species and isolates are not presented in a coherent order, making it challenging for readers to follow. For instance, there are 18 different *F. nirenbergiae* isolates scattered throughout the table. Furthermore, certain fungal species, such as *F. pharetrum* (CPC 30822 and CPC 30824), *F. oxysporum* (CBS 144134 and CBS 144135), and *F. veterinarianium* isolates (NRRL 62542, 62545, and 62547), are repetitively listed. To enhance the table's clarity, it is advisable to organize the information in a more systematic manner, such as grouping isolates by species and providing proper citations for all data derived from Lombard et al. This will facilitate a more effective and reader-friendly presentation.

Response: We greatly appreciate your suggestions. The order of the species or isolates in Table 3 was reorganized and repetitive isolates were deleted. Sequence information primarily sourced from

Lombard et al has been indicated in the footnote under Table 3.

Comment #3: The description of the phylogenetic analysis in section 3.1 appears to be confusing. The authors mentioned that they obtained *tef1*, *rpb2*, *tub2*, and *cmdA* sequences from 34 representative isolates of FOOSC from Jilin. However, in Figure 1, only seven of these isolates were analyzed. It would be helpful for the authors to clarify why only seven isolates were included in the analysis and whether the sequence information for all 34 isolates is presented elsewhere in the study.

Additionally, there seems to be a discrepancy in the total number of sequences compared. The text mentions 34 FOOSC isolates and 91 other *Fusarium* strains, totaling 125 isolates or strains. However, the statement that a total of 360 sequences were compared is inconsistent with the initial count. The authors should provide clarification on how they arrived at the total of 360 sequences and reconcile it with the initially mentioned 125 isolates or strains. Why is it necessary to analyze many other *Fusarium* species, like *F. nirenbergiae* and *F. veterinarianium*, but not out group species, like *F. solani*, *F. graminearum*, and *F. fujikuroi*?

As the authors mentioned that "The topology of the phylogenetic tree constructed based 198 on the *tef1* sequence of 132 isolates was almost the same as that of the phylogenetic tree constructed based on the 199 sequences of the *tef1*, *rpb2*, *tub2*, and *cmdA* genes.", then why is it necessary to have Figure 2?

Response: We are sorry for these errors. The 34 isolates included seven strains from *Fusarium tobaccum* sp. nov., and other 27 strains from other three species namely *F. cugenangense*, *F. odoratissimum* and *F. foetens*. In Figure 1, the total number of strains involved should be 98, including 7 strains of *F. tobaccum* sp. nov. and 91 reference strains of other FOOSC. So the total number of sequences involved should be 392, not 360. Those errors have been corrected in the manuscript.

For the selection of reference strains, *F. nirenberguae* and *F. veterinarianium* are all species of FOOSC, which are very helpful for the identification of new species of FOOSC. *F. udum* (CBS 177.31) was selected as the outgroup strain, all referring to that of Lombard *et al.* 2019 (39). The study of Lombard *et al.* has formed an epitypification of FOOSC, which is very convincing in the identification of FOOSC. Therefore, other strains and sequences were not chosen for constructing the multi-gene phylogenetic tree.

Every one of the four genes (*tef1*, *rpb2*, *tub2*, and *cmdA*) was used to construct a single-gene phylogenetic tree (not mentioned in the manuscript), and the phylogenetic tree based on *tef1* gene was

similar to that of the four-gene (*tef1*, *rpb2*, *tub2*, and *cmdA*) phylogenetic tree (The results from Lombard et al. 2019, also showed similar results). The strains of *F. tobaccum* sp. nov. were well distinguished from other strains of FOOSC by using *tef1* phylogenetic analysis, but not all of FOOSC could be distinguished, such as *F. callstephi*, *F. nirenberguae*, *F. hoodiae* and *F. oxysporum*. This study confirmed that the *tef1* gene was effective for distinguishing *F. tobaccum* sp. nov., and other 125 strains were identified as *F. tobaccum* sp. nov. based on *tef1* gene phylogenetic analysis except the 7 strains of *F. tobaccum* sp. nov. in Figure 2. Therefore, Figure 2 is necessary in the manuscript.

Comment #4: The authors should give detailed information of all statistics performed and graphs shown in Figures. What is it for the HSD test? And what do the error bars mean? In the legends (except figure 9), the authors indicate that "Bars with different uppercase or lowercase letters", but there's no uppercase letter.

Response: We greatly appreciate your suggestions. The data on biological characteristics of the new species *F. tobaccum* sp. nov. have been supplemented in Table 4 and Table 5. In addition, the test method used in this experiment is Duncan's multiple range test not HSD (Honestly Significantly Different). And data in Figures 7, 8, and 9 are analyzed by the one-way ANOVA, so there's no uppercase letter. All related errors have been revised in legends of Figure 7, 8, 9, and 10.

Comment #5: The authors conducted growth experiments on isolates; however, it is notable that there is no mention of a control species, such as the previously referenced *F. cugenangense*.

Response: Thank you for your insightful suggestion. A description of the morphological comparison between *F. tobaccum* and *F. cugenangense* and morphological photographs were added to the manuscript (Figure 4, Table 4). The effects of nutritional conditions and environmental factors on mycelial growth of *F. cugenangense* were also conducted. However, considering the overall page numbers of the manuscript, the detail data about *F. cugenangense* is not concerned much in the manuscript for the time being. If necessary, we could supplement the corresponding data.

Comment #6: Do the FOOSC isolates pathogenic to tomato as well?

Response: Thank you for your question. We inoculated with *Fusarium tobaccum* sp. nov. and *F. cugenangense* on tomato at the same time in greenhouse, the results showed that *F. tobaccum* sp. nov.

could infect tomato slightly but *F. cugenangense* could not infect tomato.

Comment #7: The authors should enhance the overall quality of their writing in English.

For example, in line 129, what is a triangle bottle?

Line 152, what are "Eight mm fungal cakes"?

Line 194, what is the abbreviation "HI" means?

Legend of Figure 3, "Reverse of colony on PDA".

Response: Thank you for your suggestion. We have invited an English native speaker to revise the manuscript. We tried our best to improve the manuscript and made some changes in the manuscript. And the changes mainly in English language are in red in the revised manuscript. These changes will not influence the content and framework of the paper.

"triangle bottle" has been revised as "conical flask".

"Eight mm fungal cakes" has been revised as "8-mm fungal plug".

"HI" means homoplasy index, it has been added in the MATERIALS AND METHODS.

"Reverse of colony on PDA" has been changed to "back of colony on PDA".

Finally, we thank the editors and the reviewers for your carefulness and patience. We hope the revised manuscript can meet the requirements for publication in this journal.

Response to the editors' and the reviewers' comments

Dear editors and reviewers,

We deeply appreciate the valuable comments and suggestions provided by you and the reviewers for our manuscript entitled "Identification and biological characteristics of *Fusarium tobaccum* sp. nov., a novel species causing tobacco root rot in Jilin Province, China" (Spectrum 00925-24). We have carefully revised the manuscript and provided a point-by-point response to the review's comments. The modifications according to the reviewers' suggestions are marked in red.

Finally, we believe the modifications have significantly improved the manuscript and thank you very much for your attention to our work. We hope we have addressed all your concerns, and please let us know if there are any further questions.

Sincerely yours,

Jie Gao, Ph. D & Professor
Jilin Agricultural University
jjegao115@126.com

Response to the comments from Reviewer #1:

1. Line 51-53: The aim of study is poorly and need to write in clear way

Response: We are sorry for the poor expression. We have rewritten this section in the manuscript.

2. the title of sections in material and methods: not sound and change it to be suitable as paper

Response: Thank you for your comments. We have revised several titles of sections in material and methods.

3. Line 56: "Diseased FWT-like samples were collected between 2018 from five towns spanning four counties or cities". Rewrite

Response: Thank you for your comments. This sentence has been rewritten according to your suggestion.

4. Change all term "Strain" to "isolate"

Response: We accept your suggestion. All term "strain" in the manuscript has been changed into "isolate".

5. What is meant = "macrospores" in line 156 and in conclusion?

Response: Asexual reproduction of *Fusarium* usually generates three types of spores, including macrospores (also named macroconidia), microspores (also named microconidia) and chlamydospores. And macrospores are significant for the pathogenicity of *Fusarium* and fungicide sensitivity. Therefore, we studied sporulation ratios of the macrospores of *F. tobaccum* sp. nov. in three liquid media. The "macrospores" mentioned in line 156 and the conclusion have been changed into macroconidia, and microspores in the text have also been changed into microconidia.

6. intercalarily in line 191: What is this? correct it

Response: Thank you for your comments. Intercalarily refers to growth or development that occurs between existing parts. In this study, "bearing terminal or intercalarily phialides" means that phialides can be produced either at the tips or in the middle of the hyphae. The use of the adverb "intercalarily" describing the morphology of *Fusarium* was referred to several related literatures. We provide several references as follows:

"Conidiophores carried on the aerial mycelium 35–75 µm tall, unbranched or sparingly branched, bearing terminal or intercalarily monophialides."

1. Lombard L, Sandoval-Denis M, Lamprecht SC, and Crous PW. 2019. Epitypification of *Fusarium oxysporum*—clearing the taxonomic chaos. *PERSOONIA*. 43:1-47.

"Chlamydoconidia abundantly formed in hyphae, globose to subglobose, (5–)14(–20) × (4–)6–12(–17) µm, formed terminally or intercalarily, single or in pairs. "

2. Maryani N, Lombard L, Poerba YS, Subandiyah S, Crous PW, and Kema GHJ. 2019. Phylogeny and genetic diversity of the banana *Fusarium* wilt pathogen *Fusarium oxysporum* f. sp. *cubense* in the Indonesian centre of origin. *STUD MYCOL*. 92: 155–194.

7. *F. cugenangense*: What is this? Why are mentioned to this species? but you took on one

Response: Thank you for your comments. In the first revision in the journal *APPLIED AND ENVIRONMENTAL MICROBIOLOGY*, Reviewer #2 had a suggestion as follows: "The authors conducted growth experiments on isolates; however, it is notable that there is no mention of a control species, such as the previously referenced *F. cugenangense*." Therefore, we accepted this suggestion and added the data of growth and morphology of *F. cugenangense*. In addition, in the phylogenetic tree constructed based on multi-genes including *tef1*, *tub2*, *cmdA*, and *rpb2*, *F. cugenangense* has the closest phylogenetic relationship with *F. tobaccum*. To demonstrate that *F. tobaccum* is a new species, we compared the morphological differences between the two species in Table 4 in the manuscript.

8. CONCLUSIONS: Poorly, and need to rewrite

Response: We apologize for our poor expressions in CONCLUSIONS section. The CONCLUSIONS has been rewritten.

9. Line 192: "polyphialidic," This characterize is not associated with *F. oxysporum* morphology

Response: Thank you for your comments. "Polyphialidic" is an essential sporulation structure in *Fusarium* species, characterized by two distinct forms: monophialidic and polyphialidic.

Response to the comments from Reviewer #2:

1. *Fusarium* wilt of tobacco root rot (FWT) or *Fusarium* wilt of tobacco (FWT)? FWT was different between ABSTRACT and IMPORTANCE.

Response: We are sorry for this error. We have changed "Fusarium wilt of tobacco root rot (FWT)" into "Fusarium wilt of tobacco (FWT)" in the ABSTRACT.

2. *Fusarium tobaccum* sp. nov. could cause tobacco root rot OR *Fusarium* wilt of tobacco? Please uniform it.

Response: We are sorry for this error. *Fusarium tobaccum* sp. nov. could cause tobacco root rot leading to Fusarium wilt of tobacco (FWT). We have uniform the term "Fusarium wilt of tobacco (FWT)" in the manuscript.

3. In line 55, "between 2018" might be error grammar. Please revise it.

Response: We are sorry for this error. We have revised this sentence.

4. In line 55, diseased samples were collected between 2018 from five towns spanning four counties or cities in Jilin. However, *Fusarium tobaccum* sp. nov. were only isolated from Tuoyaoling town, Liuhe county? Please confirm it.

Response: We apologize for the misunderstanding of the isolate numbers due to our unclear expression. A total of 132 isolates were isolated from five towns, and only 7 isolates mentioned and listed in Table 1 in the manuscript were used for the identification in this study.

5. Longitude and latitude of sampling site should be provided in Table 1.

Response: We appreciate your valuable suggestion. The information of longitude and latitude of sampling sites has been added in Table 1.

6. In Line 57, authors give this sentence "A total of 132 pure cultures were obtained and seven strains were further studied (Table 1)". However, seven strain were only presented in Table 1. Please make it clear.

Response: Thank you for your comments. We have revised the related descriptions in the manuscript. In this study, we successfully obtained 132 isolates pathogenic to tobacco and 7 isolates were chosen for subsequent research, which were identical to the remaining 125 isolates on morphological characteristics from five towns in Jilin province. If needed, we can supplement them in Table 1.

7. In Line 61, Media used in this study were provided in this section. Please give suitable References to support each Media formula. In addition, pH value is important for the culture of the tested strains. Please give pH value for Media.

Response: We appreciate your valuable suggestions. We have added the corresponding references to support each media formula.

8. In Line 134, among seven strains, authors have selected strain LH156101 for biological characteristics, why?

Response: All seven isolates of *Fusarium tobaccum* sp. nov. were almost identical on colony morphology. Therefore, we randomly chose LH156101 as the representative strain for biological characteristics study.

9. In Line 329, the sentence of "Effects of pH, temperature, media, carbon source, nitrogen source and light on the mycelial growth of *F. tobaccum* sp. nov. were determined;" is not suitable in CONCLUSIONS. Please provide the optimal conditions for *F. tobaccum* sp. nov. In addition, this sentence was repeated with the sentence in Line 320.

Response: We appreciate your valuable suggestions. We have added the optimal conditions for *F. tobaccum* sp. nov. and also revised the other sentences in CONCLUSIONS section.

Response to the editors' and the reviewers' comments

Dear editors and reviewers,

We appreciate the valuable comments and suggestions from you and the reviewers to our manuscript entitled “Identification and biological characteristics of *Fusarium tobaccum* sp. nov., a novel species causing tobacco root rot in Jilin Province, China” (ID: #AEM00060-24). We have revised the manuscript carefully and provided a point-by-point response to the review’s comments. The modifications according to the reviewers’ suggestions are highlighted in red. In addition, Tobacco Fusarium root rot (TFRR) in the manuscript has been changed to Fusarium wilt of tobacco(FWT) and highlighted in red color.

Finally, we think the modifications have improved the manuscript and thank you very much for your attention to our manuscript. Hopefully we have answered all your questions and please let us know if you have any further concerns.

Sincerely yours,

Jie Gao, Ph. D & Professor

Jilin Agricultural University

jiogao115@126.com

1. Response to comments/suggestion of the reviewer #1s' comments

The authors have made the revisions based on my previous comments and suggestions. No further comments.

Response: Thank you for your rcomments.

2. Response to comments/suggestion of the reviewer #3s' comments

The idea of the manuscript is acceptable to publish in this journal. But this study lack to accurate of explaining the results. I have several comments, as follow:

Comment #1. Title: it is unsuitable because you already identified as FOSC then you reidentified.

Response: Thank you for your suggestion. The new species identified in this study do not conflict with *Fusarium oxysporum* complex species (FOSC). With the studies on *F. oxysporum* in recent decades, *F. oxysporum* has evolved from a single species to today's FOSC. FOSC belongs to a subgenus taxon, and

there are few obvious morphological differences among strains belonging to the same complex species, which can often be identified to species by phylogenetic analysis. In this study, we used multi-locus sequence analysis of *tef1-rpb2-tub2-cmdA* and *tef1*, combined with morphological characteristics to further identify the strains of FOOSC and identify them to the taxon "species", *Fusarium tobaccum* sp. nov, which is one species of the FOOSC. Therefore, we prefer to keep the original title.

Comment #2. Line 62: "*F. cugenangense* LH41 was selected as the control strain." Why are you added this as control?

Response: Thank you for your comments. In the previous round of review, reviewer 2 suggested adding data of the species which is close to *Fusarium tobaccum* sp. nov.. *F. cugenangense* is the closest to *Fusarium tobaccum* sp. nov. in phylogenetic analysis in this study, so we added the data of *F. cugenangense* for comparison with *F. tobaccum* sp. nov. and on the basis of comparative morphology between the two species it may reveal some making *F. tobaccum* sp. nov. unique.

Comment #3. Table 1: You mention for "*Fusarium tobaccum* sp. nov. isolates" in this table but in full text mentioned for FOOSC?

Response: Thank you for your comments. I have revised them in the manuscript according to your suggestion.

Comment #4. Line 105: "According to the identification results,":

What are the conical flask contained?

Response: Thank you for your comments. I may not fully understand what you mean. Your question may be two as following. Firstly," According to the identification results "refers to the results of morphology and molecular biology, a total of 132 strains of *Fusarium tobaccum* sp. nov. were identified, and the isolation frequency of five regions was determined by their isolation locations. Secondly, you mentioned " what are the conical flask contained?" in which is the barley grain–fungus for inoculation.

Comment #5. Line 222-224:"The 132 isolates of the FOOSC" ... to ... "characteristics.":

Depending on phylogenetic analysis was 7 isolates *F. tobaccum* previously, but you mentioned 132 isolates identified as "*F. tobaccum* ".

Response: Thank you for your comments. First, out of the 132 strains, seven were identified through multi gene joint analysis of *tef1-rpb2-tub2-cmdA*, while the remaining 125 strains were distinguished through *tef1* single gene sequence phylogenetic analysis, which is effective as that of multi-locus

sequence analysis by our confirmation (Results of single-gene phylogenetic trees for *tub2*, *cmdA*, and *rpb2* respectively, are not shown in the manuscript, which is the same as the results of Lombard et al. 2019), 125 strains were clustered the same branch with 7 strains identified through multi gene analysis and *Fusarium* sp. CBS 680.89, CBS 128.81, and CBS 130323 identified by Lombard et al. using multi-locus sequence analysis. So there are a total of 132 strains were identified as a new species.

Comment #6. Conclusions: The authors must add a separate section for conclusion:

Response: Thank you for your comments. I have added conclusions in the manuscript.

CONCLUSIONS

In this study, a total of 132 FOOSC strains were identified as a new species, *F. tobaccum* sp. nov. Zhao Xie & Jie Gao, based on multi-locus sequence analysis of *tef1-rpb2-tub2-cmdA* and *tef1*, combined with morphological characteristics, which is different from other species of FOOSC in morphology of conidia and genetic characteristics.

Effects of pH, temperature, media, carbon source, nitrogen source and light on the mycelial growth of *F. tobaccum* sp. nov. were determined; PD medium was optimal for total sporulation of *F. tobaccum* sp. nov.; sporulation ratios of the macrospores of *F. tobaccum* sp. nov. in PD, SN, and CMC were relatively low.

Comment #7. The authors must morphological characterize that used to discriminate among these species of *Fusarium*

Response: Thank you for your comments. We agree with you. For most of species of fungi, morphological differences are important for the distinguishment of different species, but for the identification of species belonging to FOOSC, morphological differences are very minor, even no differences among the species of FOOSC. So it is difficult to distinguish species of FOOSC only by morphology, molecular techniques are the key way or only way for distinguishing species of FOOSC. Until Lombard et al. established epitypification of FOOSC in 2019, the species being referred to as *F. oxysporum* were distinguished as 21 different species within FOOSC by multi-locus sequence analysis described by Lombard et al.. At present, due to the development of molecular biology, the optimal method for the identification of species in FOOSC is based on molecular biology and supplemented by morphology.

3. Response to comments/suggestion of the reviewer #4s' comments

Fusarium oxysporum and *F. solani* are the dominant species causing tobacco root rot. From 132 pure cultures of *F. oxysporum* from the diseased samples, the authors designated one as a novel species, *Fusarium tobaccum* sp. nov., based on the classical morphological and molecular evidences. The authors did a lot of works and gave a positive responses to the reviewers' comments. However, It is so limited to distinguish the *F. oxysporum* species based on the morphology characteristics and several partial fragments of marker genes. It's hard to understand that the new species *Fusarium tobaccum* sp. nov. Zhao Xie & Jie Gao designated by the authors from FOOSC group is not more closed to the *Fusarium oxysporum* in the phylogenetic tree other than *F. cugenangense* LH41. Some of the morphology characteristics, such as the size of macrospore, more or less chlamydospores, etc., are not much of the qualitative traits if those still have overlapping with each other with influence of environmental conditions. Therefore, the combined multiple molecular tools seem to be more important facing the distinguishing the subtle morphological differences of the cryptic species. As to the high efficacy and resolution approach, such as inter simple sequence repeat (ISSR), universal rice primer (URP), etc, the authors did not mentioned at all. The evidence is not enough to determinate a novel species from FOOSC.

Response: Thank you for your comments. First of all, you said that “It's hard to understand that the new species *Fusarium tobaccum* sp. nov. Zhao Xie & Jie Gao designated by the authors from FOOSC group is not more closed to the *Fusarium oxysporum* in the phylogenetic tree other than *F. cugenangense* LH41”, based on our phylogenetic analysis of these three species. According to the new taxonomy of FOOSC, *F. cugenangense* and *F. oxysporum* are the species in *Fusarium oxysporum* species complex. Now the definition of “*F. oxysporum*” is only a single species of FOOSC, which is different from that of previous “*F. oxysporum*” (many species collection) before the coming up of “*Fusarium oxysporum* species complex (FOOSC)”. The above three species (*F. cugenangense*, *F. oxysporum*, and *Fusarium tobaccum* sp. nov.) are only three species belonging to FOOSC, and the genetic relationship between them is determined by four specific gene sequence analysis. And the results from our study are consistent with those of Lombard et al in 2019. Therefore, it's not hard to understand that the new species *Fusarium tobaccum* sp. nov. Zhao Xie & Jie Gao designated by the authors from FOOSC group is more closed to *F. cugenangense* LH41 in the phylogenetic tree other than the *F. oxysporum*.

Secondly, as for the morphological overlap between the morphology of the FOSC species, and there is no specific character, we agree with you on this point. However, within FOSC, it is impossible to distinguish the different species of FOSC by morphology. Before the concept of FOSC, many different species in FOSC were named as the same species *F. oxysporum*. With the development of molecular biology in recent years, great progress has been made in the identification of “*F. oxysporum*” (many species collection). Many techniques such as RAPD, SSR, ISSR, and URP were only used to genetic diversity among different strains, not for distinguishment of different species. moreover, these techniques are time-consuming and labor-intensive and difficult to distinguish all species in FOSC. In the past 30 years, researchers mainly focus on specific genes for the classification of *F. oxysporum*, so that they could applied for distinguishing the different species based on the gene sequences analysis of different species. Until 2019, Lombard et al. established epitypification of *F. oxysporum* based on sequences analysis *tef1*, *rpb2*, *tub2*, and *cmdA* genes in response to the confusion of FOSC. At present, it is commonly used for identification of different species of FOSC based on sequences analysis of *tef1*, *rpb2*, *tub2* and *cmdA* genes rather than RAPD, SSR, ISSR, etc. So we are sure that it is enough to identify *F. tobaccum* sp. nov. as a novel species in FOSC.

Minor:

Comment #1: To select the reference strains of *Fusarium* spp., why only four *F. oxysporum* spp. (fp. what) and what's the purpose not let other *F. oxysprum* formae speciales in?

Response: Thank you for your comments. Lombard et al. (2019) identified 21 species in FOSC. The purpose of this study is to distinguish the different species in FOSC causing *Fusarium* wilt of tobacco. We tried to know whether there is different species in FOSC isolates from tobacco. The reference strains selected in this study were all referred to the study of Lombard et al. 2019, formae speciales are the taxon below the species of *Fusarium* and mainly divided by pathogenicity on different host. So we did not select the strains of *F. oxysprum* formae speciales. It's worth noting that nomenclature of *Fusarium oxysporum* formae speciales are not subject to “the International Code of Nomenclature for algae, fungi, and plants”. For the FOSC, formae speciales are defined by the accessory chromosome obtained via horizontal gene transfer, and the pathotype on the type of virulence genes carried by this chromosome and should not be confused with the species boundaries within the FOSC.

Comment #2: Just applying the *tefl* sequence only, the four *F. oxysporum* spp. reference strain were not classed into a closed group. Maybe the other sequence dataset could also drive the novel species into different group.

Response: Thank you for your comments. What you said is possible. Our study tried to find methods to distinguish *Fusarium tobaccum* sp. nov. from large numbers of species in FOSC. It is true that all cryptic species are not completely separated with the *tefl* single-gene phylogenetic analysis. But we have confirmed that *tefl* single-gene phylogenetic analysis is effective as that of multi-gene phylogenetic analysis (unpublished data). Based on *tefl* single-gene phylogenetic analysis 125 FOSC strains from tobacco were completely clustered in the same branch with the 7 strains which has been identified as *F. tobaccum* sp. nov. by multi-gene phylogenetic analysis, as well as *Fusarium* sp. CBS 680.89, CBS 128.81, and CBS 130323 (three strains which are unnamed new species identified by Lombard et al.). It can be considered that these 125 FOSC strains and 7 *F. tobaccum* sp. nov. strains are the same species.

Comment #3: The symptoms of diseased tobacco plants did not show how the vesicular system became where *F. oxysprum* typically dwells in, while some of the other *Fusarium* spp. may not. If the authors do want to name the strain as a novel species from FOSC in tobacco, the comparison with the other specific species varieties in FOSC and pathogenicity test on different host might be seriously considered.

Response: Thank you for your comments. In our other study, host range experiments were carried out on 28 crop species belonging to 27 genera (unpublished data). The selected strains of *Fusarium tobaccum* sp. nov. were highly pathogenic to tobacco, and was pathogenic to varying degrees to chili, cotton, mung bean, pea, bitter gourd, peanut, and sweet potato. It did not show specificity to tobacco. For the species nomenclature, one can choose the host name from which you initially isolated, for example: nomenclature of *F. elaeidis* refers to genus name of the host plant, *Elaeis* is from which this fungus was first isolated, nomenclature of *F. fabacearum* referred to the name of plant family, Fabaceae, which includes the plant host *Glycine max*, from which this fungus was first isolated; nomenclature of *F. gossypinum* referred to the name of plant genus *Gossypium*, from which this fungus was isolated; nomenclature of *F. hoodiae* referred to the name of plant genus *Hoodia*, from which this

fungus was isolated.

Comment #4: The author also mentioned the *Fusarium oxysporum* in tobacco has a genetically diversity, classified to the *F. oxysporum* f. sp. *nicotianae* or *F. oxysporum* (Schlecht) Wr. var. *nicotianae* Johnson. What's the relationship with the novel species in this study?

Response: Thank you for your comments. It is reported that *F. oxysporum* f. sp. *nicotianae*, *F. oxysporum* f. sp. *vasinfectum*, and *F. oxysporum* f. sp. *batatas* could cause FWT, and their host ranges overlap (Johnson et al., 1921; Anderson et al., 1944), but the authors only did inoculate with the isolated *Fusarium* strains on tobacco, sweet potato and cotton, they did not inoculate *Fusarium* on other plants and did not identify the *Fusarium* fungi by multi-locus sequence analysis. This informal subspecific rank of forma specialis is defined based on the plant pathogenicity of the particular *F. oxysporum* strain (Armstrong & Armstrong 1981, Gordon & Martyn 1997, Kistler 1997, Baayen et al. 2000, Leslie & Summerell 2006). Therefore, *F. oxysporum* strains attacking the same plant host are generally considered to belonging to the same forma specialis. This homologous trait has led to erroneous assumptions considering a specific forma specialis to be phylogenetically monophyletic. According to the current methods and results that cross inoculation, host range test and phylogenetic analysis of polygalacturonase by us (unpublished data), *F. oxysporum* f. sp. *nicotianae* or *F. oxysporum* (Schlecht) Wr. var. *nicotianae* identified by Johnson et al. in 1944 may be one species of FOOSC other than forma specialis of *Fusarium oxysporum*. Whether the species is the same as *F. tobaccum* sp. nov. or other species need to be further studied, which is difficulty to obtain the strains *F. oxysporum* f. sp. *nicotianae* or *F. oxysporum* (Schlecht) Wr. var. *nicotianae* identified by Johnson et al.

Finally, we thank the editors and the reviewers for your carefulness and patience. We hope the revised manuscript can meet the requirements for publication in this journal.

Response to the editors' and the reviewers' comments

Dear editors and reviewers,

We appreciate the valuable comments and suggestions from you and the reviewers to our manuscript entitled "Identification and biological characteristics of *Fusarium tobaccum* sp. nov., a novel species causing tobacco root rot in Jilin Province, China" (formerly ID: # AEM00481-24). We are pleased to transfer this manuscript to Microbiology Spectrum (Spectrum 00925-24). We have revised the manuscript carefully and provided a point-by-point response to the review's comments. The modifications according to the reviewers' suggestions are highlighted in red.

Finally, we think the modifications have improved the manuscript and thank you very much for your attention to our manuscript. Hopefully we have answered all your questions and please let us know if you have any further concerns.

Sincerely yours,
Jie Gao, Ph. D & Professor
Jilin Agricultural University
jiegao115@126.com

1. Response to comments/suggestion of the reviewer #1s' comments

The author has responded to each comment according to the reviewer's suggestions, and added a "Conclusion" section at the end of the article. In the discussion, the authors discuss the relationship between the novel species and *F. oxysporum* f. sp. *nicotianae*. I agree with the author's suggestion to distinguish the FOOSC strains causing tobacco wilt disease into four species rather than forma specialis of *Fusarium oxysporum*.

Comment #1. Biological characteristics are important for fungi, do the authors compare the novel species with the other related species of *Fusarium*?

Response: Thank you for your comments, I wholeheartedly agree with your viewpoint. This study conducted additional experiments on the biological characteristics of other related *Fusarium* species, such as *F. cugenangense* and *F. odoratissimum*. However, due to considerations of logical coherence in the research, these experiment results were not included in the manuscript.

Comment #2. The authors identify a novel species *Fusarium tobaccum* sp. nov., what are the main differences with other closed species such as *F. cugenangense* LH41 selected by you as the control strain.

Response: Thank you for your comments, I wholeheartedly agree with your viewpoint. According to your and the fifth reviewer's opinion, we have added morphological comparison of the other two closed species (*F. callistephi* and *F. elaeidis*) with *Fusarium tobaccum* sp. nov. in the manuscript and found the uniqueness, thank you again for your comments. (Line 312-320)

Comment #3. The authors have changed the disease name tobacco *Fusarium* root rot of (TFRR) to *Fusarium* wilt of tobacco (FWT), please change Line 40 and 50 "TFRR" to "FWT".

Response: Thank you for your comments, which have been corrected in the manuscript. (Line 8, 20, 34,

40, and 50)

Comment #4. Other minor errors should be double-checked based on the journal format.

Response: Thank you for your comments, which have been corrected in the manuscript.

2. Response to comments/suggestion of the reviewer #3s' comments

Comment #1. Title: it is unsuitable because you already identified as FOOSC then you reidentified. I don't understand the answer, they already mentioned FOOSC and found 132 from 320. The researchers can not understand this, what FOOSC and then select 132.

Response: Thank you for your comments. In order to make it easier for readers to understand our research, we have tried to avoid using FOOSC in the revised manuscript. Given the considerable numbers of isolated pathogens from diseased tobacco, we firstly determined whether the strains belonging to *Fusarium oxysporum* species complex (FOOSC) through morphological and phylogenetic analysis of *tef1* genes. Only the strains were determined belonging to FOOSC, we could selected the *tef1*, *tub2*, *cmdA*, and *rpb2* genes rather than other genes to distinguish the different species of FOOSC (preliminary findings are omitted from this manuscript).

Let us explain to you the reason for this description. *Fusarium oxysporum* species complex is a taxon belonging to a subgenus that has taxonomic status below the genus *Fusarium*, it is above 30 species such as *F. oxysporum*, *F. cugenangense*, *F. odoratissimum*, *F. foetens*, *F. tardichlamydosporum*. At first, the identification of the species mainly relied on morphological differences, but with the development of molecular biology, it is possible to distinguish morphologically similar but genetically different species by using different genes, Therefore, in recent years, "*Fusarium oxysporum*" has evolved into "*Fusarium oxysporum* species complex", and new species have been continuously identified through molecular biology.

The haplotype subspecific classification system was introduced by Chang et al. (2006) and later expanded upon by O'Donnell et al. (2008, 2009) to include strains from both the FOOSC and *Neocosmospora* (formerly the *F. solani* (FSSC) species complex). This classification system is based on unique multi-locus genotypes within the species complex, aimed to resolve communication problems among public health and agricultural scientists (O'Donnell et al. 2008). Chang et al. (2006) proposed a standardised haplotype nomenclature system that depict the species complex, species and genotype. O'Donnell, Lombard et al. (2019) used phylogenetic analysis to establish Epytification of *Fusarium oxysporum* through four genes (*tef1*, *cmdA*, *tub2*, and *rpb2*). This is currently the best method to identify different species in the *Fusarium oxysporum* species complex.

Summary, "320 FOOSC" mentioned in this study refers to the 320 strains belonging to the subspecies of *Fusarium* spp. In the *Fusarium oxysporum* species complex, the specific species are unknown. Whereas "132" indicates that 132 of 320 *Fusarium* spp. strains were identified as species by phylogenetic analysis, and these 132 *Fusarium* spp. were located in a new branch in the phylogenetic tree, so in this study, these 132 *Fusarium* spp. strains were identified as a new species and named *Fusarium tobaccum* sp. nov.

References:

1. Chang DC, Grant GB, O'Donnell K, et al. 2006. Multistate outbreak of *Fusarium keratitis* associated with use of a contact lens solution. *Jama*. 296: 953–963.
2. O'Donnell K, Sutton DA, Fothergill A, et al. 2008. Molecular phylogenetic diversity,

multilocus haplotype nomenclature, and in vitro antifungal resistance within the *Fusarium solani* species complex. *J Clin Microbiol.* 46: 2477–2490.

3. O'Donnell K, Gueidan C, Sink S, et al. 2009. A two-locus DNA sequence database for typing plant and human pathogens within the *Fusarium oxysporum* species complex. *Fungal Genet Biol.* 46: 936–948.
4. Lombard L, Sandoval-Denis M, Lamprecht SC, and Crous PW. 2019. Epitypification of *Fusarium oxysporum*—clearing the taxonomic chaos. *Persoonia.* 43:1-47.

Comment #2. Line 222-224:"The 132 isolates of the FOSC" ... to ... "characteristics.": Depending on phylogenetic analysis there were 7 isolates of *F. tobaccum* previously, but you mentioned 132 isolates identified as "*F. tobaccum*". This represents a big ambiguity, and the authors are using a multi gene joint analysis for 7 isolates only but in conclusions, they mentioned using multi gene for 132.

Response: Thank you for your comments. We are so sorry for this big ambiguity. In view of your comments, we have removed 125 of 132 strains of *Fusarium* isolated from tobacco that identified based on *tef1* gene phylogenetic analysis and related parts in the revised manuscript, and only 7 strains were retained and identified using multi gene phylogenetic analysis and morphology.

Comment #3. In material and method: The authors mentioned about 132 isolates of tobacco belonged to the *F. tobaccum* sp. nov. directly after collection. This identification was depending on what? How could the researchers determine the species and gave a new species directly without any procedure before. In addition, the authors mentioned to FOSC isolates. The table contained on pathogenicity. This is ambiguous. The ideas do not flow logically.

Response: Thank you for your comments. I agree with your opinion, which is more rigorous, and I have revised it in the manuscript (Line 59). The pathogenicity mentioned in Table 1 is not irrelevant in this manuscript. The strains isolated and identified in this study need to be verified by Koch's postulates to determine that they are pathogenic to tobacco. The pathogenicity in the table is the result of Koch's postulates verification.

Comment #4. Morphological for *Fusarium tobaccum*: The authors are depending on characterizations of species not helpful to consider as new species. The authors mentioned characterizations not useful to distinct for a new species from other *Fusarium* spp.

Response: Thank you for your comments. I agree with your opinion, *Fusarium oxysporum* species complex (formerly the *F. oxysporum*) has subtle morphological differences, not only between *F. tobaccum* sp. nov. and *F. cugenangense* as mentioned in this study, it is an objective fact that the morphological differences between other species in FOSC are also very subtle. However, the identification of *F. oxysporum* species complex is mainly based on molecular biology and supplemented by morphology. In other words, without the current identification methods of molecular biology, the taxon of the subgenus *F. oxysporum* species complex would not exist, the taxon of the subgenus complex species was established because of molecular biology.

3. Response to comments/suggestion of the reviewer #5s' comments

The authors of Xie and his/her colleagues reported the identification of a novel *Fusarium* species named *Fusarium tobaccum* sp. nov., from tobacco plants causing tobacco root rot. This novel species was identified based on the classical morphological and molecular phylogenetic analysis using multi-locus sequences. The tasks associated with this manuscript are substantial and very meaningful. However, i have several comments listed as follows:

Comment #1. The identification of *Fusarium* species is one of the most critical issues in fungal taxonomy. From the phylogenetic analysis, it seems reasonable to classify the *Fusarium* fungus in this study as a novel species. However, the morphological identification needs to be strengthened, as only one strain of LH156101 has been used for cultural and morphological characterization. Usually, different strains of *Fusarium* are easily differentiated in physiological phenotypes, thus, phenotypic analysis with multiple strains is warranted.

Response: Thank you for your approval of this study and your valuable comments. For the identification of morphology, only the morphological result of strain LH156101 is presented in the manuscript. In this study, all the seven strains identified by multi-gene phylogenetic analysis were morphologically characterized and we found that the morphology of seven strains were almost the same. Therefore, only the morphological phenotypes of strain LH156101 were presented, and represented the common characteristics of *F. tobaccum* sp. nov.

Comment #2. In the phylogenetic analysis, since only the *tef1* single gene was used, i think it is arbitrarily to classify the other 125 strains as *Fusarium tobaccum* sp. nov., and that the topological structure of the monogenic phylogenetic tree was different from that of polygenic phylogenetic tree. In other words, even if the *tef1* sequence is exactly the same, it can not be classified as a specific cryptic species under the FOOSC. This can be observed by BLASTn analysis on the *Fusarium-ID* database using individual *tef1* sequence.

Response: Thank you for your comments. I wholeheartedly agree with your viewpoint. In view of reviewer 3's comments and your comments, we have removed 125 of 132 strains of *Fusarium* isolated from tobacco that identified based on *tef1* gene phylogenetic analysis and related content in the revised manuscript, and only 7 strains retained and were identified using multi gene phylogenetic analysis and morphology.

Comment #3. According to the reference Lombard et al., 2019 and the phylogenetic analysis in this study, *F. callistephi*, *F. elaeidis*, and *F. cugenangense* are probably the most closely related species of *Fusarium tobaccum* sp. nov., thus, if possible, comparison of morphological or other physiological and biochemical characteristics or genetic background of these four species would make the results of this paper more rigorous.

Response: Thank you for your comments. I wholeheartedly agree with your viewpoint. We have added morphological contrasts of four closed species to the manuscript. Because of *F. callistephi* and *F. elaeidis* were not isolated in this experiment, the morphological description of these two strains was referred to Lombard et al., 2019. (Line 312-320)

Comment #4. For pathogenicity assays, the authors should evaluate the pathogenicity differentiation according to the disease index, not just the disease grades. The authors also

did not account for whether the disease symptoms was consistent across all plants in each treatment. In figure 5, the symptoms of the four strains looks similar, but they were defined in different disease grades of 2 and 3. The symptoms of vascular discoloration should be exhibited.

Response: Thank you for your comments. The method used in this study to measure the severity of disease by mean disease grade was referenced from Clark et al., 1998. The onset symptoms presented in Figure 4 are only representative, while the average disease level in Table 1 is the result of 10 biological replicates, so there might be discrepancies between the two due to these differences. Your understanding in this regard would be greatly appreciated.

Comment #5. In Line 276, the sporulation quantity was inconsistent with that in Table 6, nor was it the same in L286.

Response: Thank you for your comments. We are so sorry for those errors. We have corrected them in the manuscript.

Comment #6. Since this disease is named "Fusarium wilt of tobacco (FWT)", this should be unified in the text.

Response: Thank you for your comments, which have been corrected in the manuscript. (Line 8, 20, 34, 40, and 50)

Re: Spectrum00925-24R1 (Identification and biological characteristics of *Fusarium tobaccum* sp. nov., a novel species causing tobacco root rot in Jilin Province, China)

Dear Prof. Jie Gao:

Thank you for the privilege of reviewing your work. Below you will find my comments, instructions from the Spectrum editorial office, and the reviewer comments.

As you will see, Reviewer 2 is satisfied with the changes made by the authors, while Reviewer 1 has some modest additional comments. In addition, there appears to be an issue with Figure 2 and 3 (which appear to be the same). Please consider the requests of Reviewer 1 and upload the correct figures.

Revision Guidelines

Sincerely,
Jonathan Snow
Editor
Microbiology Spectrum

Reviewer #1 (Comments for the Author):

The authors answered the comments but still some comments, as following:

1. Abstract:

Must mention to all morphological and molecular characteristics for new species that found in results to useful for describing this species

2.Line 55-57:"These findings" to end:
Delete it

3.CONCLUSIONS:

You must add what are you conclude from your study without repeating to results

Reviewer #2 (Comments for the Author):

The manuscript entitled "Identification and biological characteristics of *Fusarium tobaccum* sp. nov., a novel species causing tobacco root rot in Jilin Province, China" (Manuscript Number: Spectrum00925-24R1) have been reviewed. The revised manuscript have modified the problems previously, and satisfied all my earlier concerns. The innovative findings were supported the experimental data from this work.

Response to the editors' and the reviewers' comments

Dear editors and reviewers,

We appreciate the valuable comments and suggestions from you and the reviewers to our manuscript entitled “Identification and biological characteristics of *Fusarium tobaccum* sp. nov., a new species causing tobacco root rot in Jilin Province, China” (ID: #AEM01776-23). We have revised the manuscript carefully and provided a point-by-point response to the review’s comments. We have also invited an English native speaker to revise the manuscript carefully. The modifications according to the reviewers’ suggestions are highlighted in yellow. And other modifications mainly in English language are in red in the manuscript.

Finally, we think the modifications have improved the manuscript and thank you very much for your attention to our manuscript. Hopefully we have answered all your questions and please let us know if you have any further concerns.

Sincerely yours,

Jie Gao, Ph. D & Professor

Jilin Agricultural University

jjegao115@126.com

1. Response to comments/suggestion of the reviewer #1s' comments

Comment #1: Was the difference between the new species and other species?

Response: Thank you for your question. According to the phylogenetic tree based on the *tef1*, *tub2*, *cmdA*, and *rpb2* gene sequences, *F. tobaccum* sp. nov. is closely related to *F. cugenangense*. However, the two species differ in morphology. Although the aerial mycelia of both *F. tobaccum* sp. nov. and *F. cugenangense* were abundant on PDA, *F. tobaccum* sp. nov. produced more pigment on PDA medium and eventually occupied the entire Petri dish (Fig. 3), and it was capable of producing chlamydoconidia on SNA and CLA media, either singly or in tandem chains. However, *F. cugenangense* only produced a small number of chlamydoconidia after 4 weeks of culture on SNA medium. In addition, *F. tobaccum* sp. nov. produced microconidia with 0–2 septa and macroconidia with 3–5 septa compared with *F. cugenangense*, which have microconidia with 0–3 septum and macroconidia with 3–6 septa. These

findings, combined with the molecular results, indicate that the isolates derived from tobacco roots identical to *Fusarium* sp. represent *F. tobaccum* sp. nov. Zhao Xie & Jie Gao. To our knowledge, this is the first report of *F. tobaccum* causing root rot of tobacco. The difference between the *F. tobaccum* sp. nov. and *F. cugenangense* were summarized in Table 4.

Comment #2: Figure 3. "C Aerial conidia (microconidia)", please change as "C. Aerial conidia (microconidia)".

Response: Thanks for your suggestion. We have unified the format of Figure legends. Therefore, we have changed it into "(C) Aerial conidia (microconidia)".

Comment #3: In the title "3.5 Effect of temperatures, pH values, and light conditions on the growth of *F. tobaccum* sp. nov." The "sp." should not be written as italic.

Response: Thanks for your suggestion. It has been revised.

Comment #4: In the paragraph "3.7 Sporulation of *F. tobaccum* sp. nov. in PD, SN, and CMC liquid media". The word "cfu/mL", whether the "cfu" is correct?

Response: Thank you for your suggestion. The "cfu/mL" has been changed as "spores/mL" in the text.

Comment #5: In the research on the biological characteristics of the new species, only the effects of nutritional conditions and environmental factors on mycelial growth were observed, why did not the author observe the effect on sporulation of the new species?

Response: We greatly appreciate your suggestions. In nutritional conditions and environmental factors on mycelial growth experiment, we also tried to obtain the data for the effects of different environmental factor and nutritional conditions on sporulation, but the results were not repeatable. So we did not present the results. However, we compared the effect of three media, PD, SN, and CMC, on the total number of spores produced and the ratio of macrospores of this species in the part "Sporulation of *F. tobaccum* sp. nov. in PD, SN, and CMC liquid media.

Comment #6: What is the basis for the author's selection of *tef1*, *tub2*, *cmdA*, and *rpb2* gene loci to identify new species under FOOSC?

Response: Thank you for your insightful suggestion. In 2019, Lombard et al. used the *tef1*, *tub2*, *cmdA*, and *rpb2* gene to distinguish different species of *Fusarium oxysporum* species complex (FOSC). In our study, we construct the phylogenetic tree based on one of the four genes *tef1*, *tub2*, *cmdA*, and *rpb2* or the multiple *tef1-tub2-cmdA-rpb2* loci, indicating that multiple gene phylogenetic analysis is most effective for distinguishing different species of FOSC. Finally, the *tef1*, *tub2*, *cmdA*, and *rpb2* gene loci were selected to identify new species under FOSC.

Comment #7: In the pathogenicity test, the author directly used the barley-fungus inoculation method, why did not choose other methods?

Response: Thank you for your suggestion. We initially used inoculation methods of root soaking in spore suspension of *Fusarium* spp. and root injury inoculation methods, but their repeatability was not good. Later, we referenced to the method "barley seed-fungus inoculation" from Fang, S. H. *et al.* in reference (50). Our repeated experiments confirmed that the barley seed-fungus inoculation method was the most effective. Therefore, the barley seed-fungus inoculation method was used in this study.

Comment #8: Table3, *F. tobaccum* sp. nov, the full name of *Fusarium* should be written when first appearing in the table.

Response: Thank you for your suggestion. According to the reviewer #2s' comments, the order of the species or isolates in Table 3 was reorganized. So the first species *Fusarium callistephi* was written as full name of *Fusarium*.

Comment #9: Figure 8 and Figure 9. at the $P < 0.05$ level (D, E, F), the "P" should be italic.

Response: Thank you for your suggestion. We have revised them in the manuscript.

Comment #10: Figure 9. In the sentence "Effects of media and incubation time on the conidiation (A) and proportion of macrospores (B) of *F. tobaccum* sp. nov. LH156101. Bars with different uppercase letters (among media) or lowercase letters (among different time points) indicate significant differences according to HSD tests at the $P < 0.05$ level". The "*F. tobaccum*" should be italic.

Response: Thank you for your suggestion, we have changed it in the Figure 9.

Comment #11: References

1. Cole, JS., and Zvenyika, ZA. Stem and root rot of tobacco transplants in Zimbabwe caused by *Rhizoctonia solani* Kuhn and *Fusarium solani* (Mart.) Sacc. Zimbabwe Journal of Agricultural Research. 20:149-152(1982). The " Kuhn" should be " Kühn".

Response: Thank you for your suggestion, we have changed it in the text.

2. Response to comments/suggestion of the reviewer #2s' comments

Comment #1: Table 1 contains GenBank accession numbers for *tef1* sequences from various isolates, yet the authors fail to explicitly mention this in the table or corresponding text. It is crucial to clarify the nature of the provided accession numbers to enhance reader understanding. Additionally, there is an inconsistency where both isolates LH1294 and LH1297 share the same accession number, MN610673. This inconsistency is also observed in Table 3. The authors should address this issue by providing unique accession numbers for each isolate or offering a clear explanation for the shared accession number. Improving the presentation of this information will contribute to the overall clarity and reliability of the manuscript.

Response: We are sorry for these errors. We've supplemented the corresponding descriptions, and changed the accession number OM162144 for the strain LH1294 and the accession number MN610673 for the strain LH1297.

Comment #2: The presentation of Table 3 in the manuscript requires improvement. A notable concern is the lack of citations for information primarily sourced from Lombard et al. Additionally, the fungal species and isolates are not presented in a coherent order, making it challenging for readers to follow. For instance, there are 18 different *F. nirenbergiae* isolates scattered throughout the table. Furthermore, certain fungal species, such as *F. pharetrum* (CPC 30822 and CPC 30824), *F. oxysporum* (CBS 144134 and CBS 144135), and *F. veterinarium* isolates (NRRL 62542, 62545, and 62547), are repetitively listed. To enhance the table's clarity, it is advisable to organize the information in a more systematic manner, such as grouping isolates by species and providing proper citations for all data derived from Lombard et al. This will facilitate a more effective and reader-friendly presentation.

Response: We greatly appreciate your suggestions. The order of the species or isolates in Table 3 was reorganized and repetitive isolates were deleted. Sequence information primarily sourced from

Lombard et al has been indicated in the footnote under Table 3.

Comment #3: The description of the phylogenetic analysis in section 3.1 appears to be confusing. The authors mentioned that they obtained *tef1*, *rpb2*, *tub2*, and *cmdA* sequences from 34 representative isolates of FOOSC from Jilin. However, in Figure 1, only seven of these isolates were analyzed. It would be helpful for the authors to clarify why only seven isolates were included in the analysis and whether the sequence information for all 34 isolates is presented elsewhere in the study.

Additionally, there seems to be a discrepancy in the total number of sequences compared. The text mentions 34 FOOSC isolates and 91 other *Fusarium* strains, totaling 125 isolates or strains. However, the statement that a total of 360 sequences were compared is inconsistent with the initial count. The authors should provide clarification on how they arrived at the total of 360 sequences and reconcile it with the initially mentioned 125 isolates or strains. Why is it necessary to analyze many other *Fusarium* species, like *F. nirenbergiae* and *F. veterinarianium*, but not out group species, like *F. solani*, *F. graminearum*, and *F. fujikuroi*?

As the authors mentioned that "The topology of the phylogenetic tree constructed based 198 on the *tef1* sequence of 132 isolates was almost the same as that of the phylogenetic tree constructed based on the 199 sequences of the *tef1*, *rpb2*, *tub2*, and *cmdA* genes.", then why is it necessary to have Figure 2?

Response: We are sorry for these errors. The 34 isolates included seven strains from *Fusarium tobaccum* sp. nov., and other 27 strains from other three species namely *F. cugenangense*, *F. odoratissimum* and *F. foetens*. In Figure 1, the total number of strains involved should be 98, including 7 strains of *F. tobaccum* sp. nov. and 91 reference strains of other FOOSC. So the total number of sequences involved should be 392, not 360. Those errors have been corrected in the manuscript.

For the selection of reference strains, *F. nirenberguae* and *F. veterinarianium* are all species of FOOSC, which are very helpful for the identification of new species of FOOSC. *F. udum* (CBS 177.31) was selected as the outgroup strain, all referring to that of Lombard *et al.* 2019 (39). The study of Lombard *et al.* has formed an epitypification of FOOSC, which is very convincing in the identification of FOOSC. Therefore, other strains and sequences were not chosen for constructing the multi-gene phylogenetic tree.

Every one of the four genes (*tef1*, *rpb2*, *tub2*, and *cmdA*) was used to construct a single-gene phylogenetic tree (not mentioned in the manuscript), and the phylogenetic tree based on *tef1* gene was

similar to that of the four-gene (*tef1*, *rpb2*, *tub2*, and *cmdA*) phylogenetic tree (The results from Lombard et al. 2019, also showed similar results). The strains of *F. tobaccum* sp. nov. were well distinguished from other strains of FOOSC by using *tef1* phylogenetic analysis, but not all of FOOSC could be distinguished, such as *F. callstephi*, *F. nirenberguae*, *F. hoodiae* and *F. oxysporum*. This study confirmed that the *tef1* gene was effective for distinguishing *F. tobaccum* sp. nov., and other 125 strains were identified as *F. tobaccum* sp. nov. based on *tef1* gene phylogenetic analysis except the 7 strains of *F. tobaccum* sp. nov. in Figure 2. Therefore, Figure 2 is necessary in the manuscript.

Comment #4: The authors should give detailed information of all statistics performed and graphs shown in Figures. What is it for the HSD test? And what do the error bars mean? In the legends (except figure 9), the authors indicate that "Bars with different uppercase or lowercase letters", but there's no uppercase letter.

Response: We greatly appreciate your suggestions. The data on biological characteristics of the new species *F. tobaccum* sp. nov. have been supplemented in Table 4 and Table 5. In addition, the test method used in this experiment is Duncan's multiple range test not HSD (Honestly Significantly Different). And data in Figures 7, 8, and 9 are analyzed by the one-way ANOVA, so there's no uppercase letter. All related errors have been revised in legends of Figure 7, 8, 9, and 10.

Comment #5: The authors conducted growth experiments on isolates; however, it is notable that there is no mention of a control species, such as the previously referenced *F. cugenangense*.

Response: Thank you for your insightful suggestion. A description of the morphological comparison between *F. tobaccum* and *F. cugenangense* and morphological photographs were added to the manuscript (Figure 4, Table 4). The effects of nutritional conditions and environmental factors on mycelial growth of *F. cugenangense* were also conducted. However, considering the overall page numbers of the manuscript, the detail data about *F. cugenangense* is not concerned much in the manuscript for the time being. If necessary, we could supplement the corresponding data.

Comment #6: Do the FOOSC isolates pathogenic to tomato as well?

Response: Thank you for your question. We inoculated with *Fusarium tobaccum* sp. nov. and *F. cugenangense* on tomato at the same time in greenhouse, the results showed that *F. tobaccum* sp. nov.

could infect tomato slightly but *F. cugenangense* could not infect tomato.

Comment #7: The authors should enhance the overall quality of their writing in English.

For example, in line 129, what is a triangle bottle?

Line 152, what are "Eight mm fungal cakes"?

Line 194, what is the abbreviation "HI" means?

Legend of Figure 3, "Reverse of colony on PDA".

Response: Thank you for your suggestion. We have invited an English native speaker to revise the manuscript. We tried our best to improve the manuscript and made some changes in the manuscript. And the changes mainly in English language are in red in the revised manuscript. These changes will not influence the content and framework of the paper.

"triangle bottle" has been revised as "conical flask".

"Eight mm fungal cakes" has been revised as "8-mm fungal plug".

"HI" means homoplasy index, it has been added in the MATERIALS AND METHODS.

"Reverse of colony on PDA" has been changed to "back of colony on PDA".

Finally, we thank the editors and the reviewers for your carefulness and patience. We hope the revised manuscript can meet the requirements for publication in this journal.

Response to the editors' and the reviewers' comments

Dear editors and reviewers,

We deeply appreciate the valuable comments and suggestions provided by you and the reviewers for our manuscript entitled "Identification and biological characteristics of *Fusarium tobaccum* sp. nov., a novel species causing tobacco root rot in Jilin Province, China" (Spectrum 00925-24). We have carefully revised the manuscript and provided a point-by-point response to the review's comments. The modifications according to the reviewers' suggestions are marked in red.

Finally, we believe the modifications have significantly improved the manuscript and thank you very much for your attention to our work. We hope we have addressed all your concerns, and please let us know if there are any further questions.

Sincerely yours,

Jie Gao, Ph. D & Professor
Jilin Agricultural University
jjegao115@126.com

Response to the comments from Reviewer #1:

1. Line 51-53: The aim of study is poorly and need to write in clear way

Response: We are sorry for the poor expression. We have rewritten this section in the manuscript.

2. the title of sections in material and methods: not sound and change it to be suitable as paper

Response: Thank you for your comments. We have revised several titles of sections in material and methods.

3. Line 56: "Diseased FWT-like samples were collected between 2018 from five towns spanning four counties or cities". Rewrite

Response: Thank you for your comments. This sentence has been rewritten according to your suggestion.

4. Change all term "Strain" to "isolate"

Response: We accept your suggestion. All term "strain" in the manuscript has been changed into "isolate".

5. What is meant = "macrospores" in line 156 and in conclusion?

Response: Asexual reproduction of *Fusarium* usually generates three types of spores, including macrospores (also named macroconidia), microspores (also named microconidia) and chlamydospores. And macrospores are significant for the pathogenicity of *Fusarium* and fungicide sensitivity. Therefore, we studied sporulation ratios of the macrospores of *F. tobaccum* sp. nov. in three liquid media. The "macrospores" mentioned in line 156 and the conclusion have been changed into macroconidia, and microspores in the text have also been changed into microconidia.

6. intercalarily in line 191: What is this? correct it

Response: Thank you for your comments. Intercalarily refers to growth or development that occurs between existing parts. In this study, "bearing terminal or intercalarily phialides" means that phialides can be produced either at the tips or in the middle of the hyphae. The use of the adverb "intercalarily" describing the morphology of *Fusarium* was referred to several related literatures. We provide several references as follows:

"Conidiophores carried on the aerial mycelium 35–75 µm tall, unbranched or sparingly branched, bearing terminal or intercalarily monophialides."

1. Lombard L, Sandoval-Denis M, Lamprecht SC, and Crous PW. 2019. Epitypification of *Fusarium oxysporum*—clearing the taxonomic chaos. *PERSOONIA*. 43:1-47.

"Chlamydoconidia abundantly formed in hyphae, globose to subglobose, (5–)14(–20) × (4–)6–12(–17) µm, formed terminally or intercalarily, single or in pairs. "

2. Maryani N, Lombard L, Poerba YS, Subandiyah S, Crous PW, and Kema GHJ. 2019. Phylogeny and genetic diversity of the banana *Fusarium* wilt pathogen *Fusarium oxysporum* f. sp. *cubense* in the Indonesian centre of origin. *STUD MYCOL*. 92: 155–194.

7. *F. cugenangense*: What is this? Why are mentioned to this species? but you took on one

Response: Thank you for your comments. In the first revision in the journal *APPLIED AND ENVIRONMENTAL MICROBIOLOGY*, Reviewer #2 had a suggestion as follows: "The authors conducted growth experiments on isolates; however, it is notable that there is no mention of a control species, such as the previously referenced *F. cugenangense*." Therefore, we accepted this suggestion and added the data of growth and morphology of *F. cugenangense*. In addition, in the phylogenetic tree constructed based on multi-genes including *tef1*, *tub2*, *cmdA*, and *rpb2*, *F. cugenangense* has the closest phylogenetic relationship with *F. tobaccum*. To demonstrate that *F. tobaccum* is a new species, we compared the morphological differences between the two species in Table 4 in the manuscript.

8. CONCLUSIONS: Poorly, and need to rewrite

Response: We apologize for our poor expressions in CONCLUSIONS section. The CONCLUSIONS has been rewritten.

9. Line 192: "polyphialidic," This characterize is not associated with *F. oxysporum* morphology

Response: Thank you for your comments. "Polyphialidic" is an essential sporulation structure in *Fusarium* species, characterized by two distinct forms: monophialidic and polyphialidic.

Response to the comments from Reviewer #2:

1. *Fusarium* wilt of tobacco root rot (FWT) or *Fusarium* wilt of tobacco (FWT)? FWT was different between ABSTRACT and IMPORTANCE.

Response: We are sorry for this error. We have changed "Fusarium wilt of tobacco root rot (FWT)" into "Fusarium wilt of tobacco (FWT)" in the ABSTRACT.

2. *Fusarium tobaccum* sp. nov. could cause tobacco root rot OR *Fusarium* wilt of tobacco? Please uniform it.

Response: We are sorry for this error. *Fusarium tobaccum* sp. nov. could cause tobacco root rot leading to Fusarium wilt of tobacco (FWT). We have uniform the term "Fusarium wilt of tobacco (FWT)" in the manuscript.

3. In line 55, "between 2018" might be error grammar. Please revise it.

Response: We are sorry for this error. We have revised this sentence.

4. In line 55, diseased samples were collected between 2018 from five towns spanning four counties or cities in Jilin. However, *Fusarium tobaccum* sp. nov. were only isolated from Tuoyaoling town, Liuhe county? Please confirm it.

Response: We apologize for the misunderstanding of the isolate numbers due to our unclear expression. A total of 132 isolates were isolated from five towns, and only 7 isolates mentioned and listed in Table 1 in the manuscript were used for the identification in this study.

5. Longitude and latitude of sampling site should be provided in Table 1.

Response: We appreciate your valuable suggestion. The information of longitude and latitude of sampling sites has been added in Table 1.

6. In Line 57, authors give this sentence "A total of 132 pure cultures were obtained and seven strains were further studied (Table 1)". However, seven strain were only presented in Table 1. Please make it clear.

Response: Thank you for your comments. We have revised the related descriptions in the manuscript. In this study, we successfully obtained 132 isolates pathogenic to tobacco and 7 isolates were chosen for subsequent research, which were identical to the remaining 125 isolates on morphological characteristics from five towns in Jilin province. If needed, we can supplement them in Table 1.

7. In Line 61, Media used in this study were provided in this section. Please give suitable References to support each Media formula. In addition, pH value is important for the culture of the tested strains. Please give pH value for Media.

Response: We appreciate your valuable suggestions. We have added the corresponding references to support each media formula.

8. In Line 134, among seven strains, authors have selected strain LH156101 for biological characteristics, why?

Response: All seven isolates of *Fusarium tobaccum* sp. nov. were almost identical on colony morphology. Therefore, we randomly chose LH156101 as the representative strain for biological characteristics study.

9. In Line 329, the sentence of "Effects of pH, temperature, media, carbon source, nitrogen source and light on the mycelial growth of *F. tobaccum* sp. nov. were determined;" is not suitable in CONCLUSIONS. Please provide the optimal conditions for *F. tobaccum* sp. nov. In addition, this sentence was repeated with the sentence in Line 320.

Response: We appreciate your valuable suggestions. We have added the optimal conditions for *F. tobaccum* sp. nov. and also revised the other sentences in CONCLUSIONS section.

Response to the editors' and the reviewers' comments

Dear editors and reviewers,

We appreciate the valuable comments and suggestions from you and the reviewers to our manuscript entitled “Identification and biological characteristics of *Fusarium tobaccum* sp. nov., a novel species causing tobacco root rot in Jilin Province, China” (ID: #AEM00060-24). We have revised the manuscript carefully and provided a point-by-point response to the review’s comments. The modifications according to the reviewers’ suggestions are highlighted in red. In addition, Tobacco Fusarium root rot (TFRR) in the manuscript has been changed to Fusarium wilt of tobacco(FWT) and highlighted in red color.

Finally, we think the modifications have improved the manuscript and thank you very much for your attention to our manuscript. Hopefully we have answered all your questions and please let us know if you have any further concerns.

Sincerely yours,

Jie Gao, Ph. D & Professor

Jilin Agricultural University

jiogao115@126.com

1. Response to comments/suggestion of the reviewer #1s' comments

The authors have made the revisions based on my previous comments and suggestions. No further comments.

Response: Thank you for your rcomments.

2. Response to comments/suggestion of the reviewer #3s' comments

The idea of the manuscript is acceptable to publish in this journal. But this study lack to accurate of explaining the results. I have several comments, as follow:

Comment #1. Title: it is unsuitable because you already identified as FOSC then you reidentified.

Response: Thank you for your suggestion. The new species identified in this study do not conflict with *Fusarium oxysporum* complex species (FOSC). With the studies on *F. oxysporum* in recent decades, *F. oxysporum* has evolved from a single species to today's FOSC. FOSC belongs to a subgenus taxon, and

there are few obvious morphological differences among strains belonging to the same complex species, which can often be identified to species by phylogenetic analysis. In this study, we used multi-locus sequence analysis of *tef1-rpb2-tub2-cmdA* and *tef1*, combined with morphological characteristics to further identify the strains of FOOSC and identify them to the taxon "species", *Fusarium tobaccum* sp. nov, which is one species of the FOOSC. Therefore, we prefer to keep the original title.

Comment #2. Line 62: "*F. cugenangense* LH41 was selected as the control strain." Why are you added this as control?

Response: Thank you for your comments. In the previous round of review, reviewer 2 suggested adding data of the species which is close to *Fusarium tobaccum* sp. nov.. *F. cugenangense* is the closest to *Fusarium tobaccum* sp. nov. in phylogenetic analysis in this study, so we added the data of *F. cugenangense* for comparison with *F. tobaccum* sp. nov. and on the basis of comparative morphology between the two species it may reveal some making *F. tobaccum* sp. nov. unique.

Comment #3. Table 1: You mention for "*Fusarium tobaccum* sp. nov. isolates" in this table but in full text mentioned for FOOSC?

Response: Thank you for your comments. I have revised them in the manuscript according to your suggestion.

Comment #4. Line 105: "According to the identification results,":

What are the conical flask contained?

Response: Thank you for your comments. I may not fully understand what you mean. Your question may be two as following. Firstly," According to the identification results "refers to the results of morphology and molecular biology, a total of 132 strains of *Fusarium tobaccum* sp. nov. were identified, and the isolation frequency of five regions was determined by their isolation locations. Secondly, you mentioned " what are the conical flask contained?" in which is the barley grain–fungus for inoculation.

Comment #5. Line 222-224:"The 132 isolates of the FOOSC" ... to ... "characteristics.":

Depending on phylogenetic analysis was 7 isolates *F. tobaccum* previously, but you mentioned 132 isolates identified as "*F. tobaccum* ".

Response: Thank you for your comments. First, out of the 132 strains, seven were identified through multi gene joint analysis of *tef1-rpb2-tub2-cmdA*, while the remaining 125 strains were distinguished through *tef1* single gene sequence phylogenetic analysis, which is effective as that of multi-locus

sequence analysis by our confirmation (Results of single-gene phylogenetic trees for *tub2*, *cmdA*, and *rpb2* respectively, are not shown in the manuscript, which is the same as the results of Lombard et al. 2019), 125 strains were clustered the same branch with 7 strains identified through multi gene analysis and *Fusarium* sp. CBS 680.89, CBS 128.81, and CBS 130323 identified by Lombard et al. using multi-locus sequence analysis. So there are a total of 132 strains were identified as a new species.

Comment #6. Conclusions: The authors must add a separate section for conclusion:

Response: Thank you for your comments. I have added conclusions in the manuscript.

CONCLUSIONS

In this study, a total of 132 FOOSC strains were identified as a new species, *F. tobaccum* sp. nov. Zhao Xie & Jie Gao, based on multi-locus sequence analysis of *tef1-rpb2-tub2-cmdA* and *tef1*, combined with morphological characteristics, which is different from other species of FOOSC in morphology of conidia and genetic characteristics.

Effects of pH, temperature, media, carbon source, nitrogen source and light on the mycelial growth of *F. tobaccum* sp. nov. were determined; PD medium was optimal for total sporulation of *F. tobaccum* sp. nov.; sporulation ratios of the macrospores of *F. tobaccum* sp. nov. in PD, SN, and CMC were relatively low.

Comment #7. The authors must morphological characterize that used to discriminate among these species of *Fusarium*

Response: Thank you for your comments. We agree with you. For most of species of fungi, morphological differences are important for the distinguishment of different species, but for the identification of species belonging to FOOSC, morphological differences are very minor, even no differences among the species of FOOSC. So it is difficult to distinguish species of FOOSC only by morphology, molecular techniques are the key way or only way for distinguishing species of FOOSC. Until Lombard et al. established epitypification of FOOSC in 2019, the species being referred to as *F. oxysporum* were distinguished as 21 different species within FOOSC by multi-locus sequence analysis described by Lombard et al.. At present, due to the development of molecular biology, the optimal method for the identification of species in FOOSC is based on molecular biology and supplemented by morphology.

3. Response to comments/suggestion of the reviewer #4s' comments

Fusarium oxysporum and *F. solani* are the dominant species causing tobacco root rot. From 132 pure cultures of *F. oxysporum* from the diseased samples, the authors designated one as a novel species, *Fusarium tobaccum* sp. nov., based on the classical morphological and molecular evidences. The authors did a lot of works and gave a positive responses to the reviewers' comments. However, It is so limited to distinguish the *F. oxysporum* species based on the morphology characteristics and several partial fragments of marker genes. It's hard to understand that the new species *Fusarium tobaccum* sp. nov. Zhao Xie & Jie Gao designated by the authors from FOOSC group is not more closed to the *Fusarium oxysporum* in the phylogenetic tree other than *F. cugenangense* LH41. Some of the morphology characteristics, such as the size of macrospore, more or less chlamydospores, etc., are not much of the qualitative traits if those still have overlapping with each other with influence of environmental conditions. Therefore, the combined multiple molecular tools seem to be more important facing the distinguishing the subtle morphological differences of the cryptic species. As to the high efficacy and resolution approach, such as inter simple sequence repeat (ISSR), universal rice primer (URP), etc, the authors did not mentioned at all. The evidence is not enough to determinate a novel species from FOOSC.

Response: Thank you for your comments. First of all, you said that “It's hard to understand that the new species *Fusarium tobaccum* sp. nov. Zhao Xie & Jie Gao designated by the authors from FOOSC group is not more closed to the *Fusarium oxysporum* in the phylogenetic tree other than *F. cugenangense* LH41”, based on our phylogenetic analysis of these three species. According to the new taxonomy of FOOSC, *F. cugenangense* and *F. oxysporum* are the species in *Fusarium oxysporum* species complex. Now the definition of “*F. oxysporum*” is only a single species of FOOSC, which is different from that of previous “*F. oxysporum*” (many species collection) before the coming up of “*Fusarium oxysporum* species complex (FOOSC)”. The above three species (*F. cugenangense*, *F. oxysporum*, and *Fusarium tobaccum* sp. nov.) are only three species belonging to FOOSC, and the genetic relationship between them is determined by four specific gene sequence analysis. And the results from our study are consistent with those of Lombard et al in 2019. Therefore, it's not hard to understand that the new species *Fusarium tobaccum* sp. nov. Zhao Xie & Jie Gao designated by the authors from FOOSC group is more closed to *F. cugenangense* LH41 in the phylogenetic tree other than the *F. oxysporum*.

Secondly, as for the morphological overlap between the morphology of the FOSC species, and there is no specific character, we agree with you on this point. However, within FOSC, it is impossible to distinguish the different species of FOSC by morphology. Before the concept of FOSC, many different species in FOSC were named as the same species *F. oxysporum*. With the development of molecular biology in recent years, great progress has been made in the identification of “*F. oxysporum*” (many species collection). Many techniques such as RAPD, SSR, ISSR, and URP were only used to genetic diversity among different strains, not for distinguishment of different species. moreover, these techniques are time-consuming and labor-intensive and difficult to distinguish all species in FOSC. In the past 30 years, researchers mainly focus on specific genes for the classification of *F. oxysporum*, so that they could applied for distinguishing the different species based on the gene sequences analysis of different species. Until 2019, Lombard et al. established epitypification of *F. oxysporum* based on sequences analysis *tef1*, *rpb2*, *tub2*, and *cmdA* genes in response to the confusion of FOSC. At present, it is commonly used for identification of different species of FOSC based on sequences analysis of *tef1*, *rpb2*, *tub2* and *cmdA* genes rather than RAPD, SSR, ISSR, etc. So we are sure that it is enough to identify *F. tobaccum* sp. nov. as a novel species in FOSC.

Minor:

Comment #1: To select the reference strains of *Fusarium* spp., why only four *F. oxysporum* spp. (fp. what) and what's the purpose not let other *F. oxysprum* formae speciales in?

Response: Thank you for your comments. Lombard et al. (2019) identified 21 species in FOSC. The purpose of this study is to distinguish the different species in FOSC causing *Fusarium* wilt of tobacco. We tried to know whether there is different species in FOSC isolates from tobacco. The reference strains selected in this study were all referred to the study of Lombard et al. 2019, formae speciales are the taxon below the species of *Fusarium* and mainly divided by pathogenicity on different host. So we did not select the strains of *F. oxysprum* formae speciales. It's worth noting that nomenclature of *Fusarium oxysporum* formae speciales are not subject to “the International Code of Nomenclature for algae, fungi, and plants”. For the FOSC, formae speciales are defined by the accessory chromosome obtained via horizontal gene transfer, and the pathotype on the type of virulence genes carried by this chromosome and should not be confused with the species boundaries within the FOSC.

Comment #2: Just applying the *tefl* sequence only, the four *F. oxysporum* spp. reference strain were not classed into a closed group. Maybe the other sequence dataset could also drive the novel species into different group.

Response: Thank you for your comments. What you said is possible. Our study tried to find methods to distinguish *Fusarium tobaccum* sp. nov. from large numbers of species in FOSC. It is true that all cryptic species are not completely separated with the *tefl* single-gene phylogenetic analysis. But we have confirmed that *tefl* single-gene phylogenetic analysis is effective as that of multi-gene phylogenetic analysis (unpublished data). Based on *tefl* single-gene phylogenetic analysis 125 FOSC strains from tobacco were completely clustered in the same branch with the 7 strains which has been identified as *F. tobaccum* sp. nov. by multi-gene phylogenetic analysis, as well as *Fusarium* sp. CBS 680.89, CBS 128.81, and CBS 130323 (three strains which are unnamed new species identified by Lombard et al.). It can be considered that these 125 FOSC strains and 7 *F. tobaccum* sp. nov. strains are the same species.

Comment #3: The symptoms of diseased tobacco plants did not show how the vesicular system became where *F. oxysprum* typically dwells in, while some of the other *Fusarium* spp. may not. If the authors do want to name the strain as a novel species from FOSC in tobacco, the comparison with the other specific species varieties in FOSC and pathogenicity test on different host might be seriously considered.

Response: Thank you for your comments. In our other study, host range experiments were carried out on 28 crop species belonging to 27 genera (unpublished data). The selected strains of *Fusarium tobaccum* sp. nov. were highly pathogenic to tobacco, and was pathogenic to varying degrees to chili, cotton, mung bean, pea, bitter gourd, peanut, and sweet potato. It did not show specificity to tobacco. For the species nomenclature, one can choose the host name from which you initially isolated, for example: nomenclature of *F. elaeidis* refers to genus name of the host plant, *Elaeis* is from which this fungus was first isolated, nomenclature of *F. fabacearum* referred to the name of plant family, Fabaceae, which includes the plant host *Glycine max*, from which this fungus was first isolated; nomenclature of *F. gossypinum* referred to the name of plant genus *Gossypium*, from which this fungus was isolated; nomenclature of *F. hoodiae* referred to the name of plant genus *Hoodia*, from which this

fungus was isolated.

Comment #4: The author also mentioned the *Fusarium oxysporum* in tobacco has a genetically diversity, classified to the *F. oxysporum* f. sp. *nicotianae* or *F. oxysporum* (Schlecht) Wr. var. *nicotianae* Johnson. What's the relationship with the novel species in this study?

Response: Thank you for your comments. It is reported that *F. oxysporum* f. sp. *nicotianae*, *F. oxysporum* f. sp. *vasinfectum*, and *F. oxysporum* f. sp. *batatas* could cause FWT, and their host ranges overlap (Johnson et al., 1921; Anderson et al., 1944), but the authors only did inoculate with the isolated *Fusarium* strains on tobacco, sweet potato and cotton, they did not inoculate *Fusarium* on other plants and did not identify the *Fusarium* fungi by multi-locus sequence analysis. This informal subspecific rank of forma specialis is defined based on the plant pathogenicity of the particular *F. oxysporum* strain (Armstrong & Armstrong 1981, Gordon & Martyn 1997, Kistler 1997, Baayen et al. 2000, Leslie & Summerell 2006). Therefore, *F. oxysporum* strains attacking the same plant host are generally considered to belonging to the same forma specialis. This homologous trait has led to erroneous assumptions considering a specific forma specialis to be phylogenetically monophyletic. According to the current methods and results that cross inoculation, host range test and phylogenetic analysis of polygalacturonase by us (unpublished data), *F. oxysporum* f. sp. *nicotianae* or *F. oxysporum* (Schlecht) Wr. var. *nicotianae* identified by Johnson et al. in 1944 may be one species of FOOSC other than forma specialis of *Fusarium oxysporum*. Whether the species is the same as *F. tobaccum* sp. nov. or other species need to be further studied, which is difficulty to obtain the strains *F. oxysporum* f. sp. *nicotianae* or *F. oxysporum* (Schlecht) Wr. var. *nicotianae* identified by Johnson et al.

Finally, we thank the editors and the reviewers for your carefulness and patience. We hope the revised manuscript can meet the requirements for publication in this journal.

Response to the editors' and the reviewers' comments

Dear editors and reviewers,

We appreciate the valuable comments and suggestions from you and the reviewers to our manuscript entitled "Identification and biological characteristics of *Fusarium tobaccum* sp. nov., a novel species causing tobacco root rot in Jilin Province, China" (formerly ID: # AEM00481-24). We are pleased to transfer this manuscript to Microbiology Spectrum (Spectrum 00925-24). We have revised the manuscript carefully and provided a point-by-point response to the review's comments. The modifications according to the reviewers' suggestions are highlighted in red.

Finally, we think the modifications have improved the manuscript and thank you very much for your attention to our manuscript. Hopefully we have answered all your questions and please let us know if you have any further concerns.

Sincerely yours,
Jie Gao, Ph. D & Professor
Jilin Agricultural University
jiegao115@126.com

1. Response to comments/suggestion of the reviewer #1s' comments

The author has responded to each comment according to the reviewer's suggestions, and added a "Conclusion" section at the end of the article. In the discussion, the authors discuss the relationship between the novel species and *F. oxysporum* f. sp. *nicotianae*. I agree with the author's suggestion to distinguish the FOOSC strains causing tobacco wilt disease into four species rather than forma specialis of *Fusarium oxysporum*.

Comment #1. Biological characteristics are important for fungi, do the authors compare the novel species with the other related species of *Fusarium*?

Response: Thank you for your comments, I wholeheartedly agree with your viewpoint. This study conducted additional experiments on the biological characteristics of other related *Fusarium* species, such as *F. cugenangense* and *F. odoratissimum*. However, due to considerations of logical coherence in the research, these experiment results were not included in the manuscript.

Comment #2. The authors identify a novel species *Fusarium tobaccum* sp. nov., what are the main differences with other closed species such as *F. cugenangense* LH41 selected by you as the control strain.

Response: Thank you for your comments, I wholeheartedly agree with your viewpoint. According to your and the fifth reviewer's opinion, we have added morphological comparison of the other two closed species (*F. callistephi* and *F. elaeidis*) with *Fusarium tobaccum* sp. nov. in the manuscript and found the uniqueness, thank you again for your comments. (Line 312-320)

Comment #3. The authors have changed the disease name tobacco *Fusarium* root rot of (TFRR) to *Fusarium* wilt of tobacco (FWT), please change Line 40 and 50 "TFRR" to "FWT".

Response: Thank you for your comments, which have been corrected in the manuscript. (Line 8, 20, 34,

40, and 50)

Comment #4. Other minor errors should be double-checked based on the journal format.

Response: Thank you for your comments, which have been corrected in the manuscript.

2. Response to comments/suggestion of the reviewer #3s' comments

Comment #1. Title: it is unsuitable because you already identified as FOOSC then you reidentified. I don't understand the answer, they already mentioned FOOSC and found 132 from 320. The researchers can not understand this, what FOOSC and then select 132.

Response: Thank you for your comments. In order to make it easier for readers to understand our research, we have tried to avoid using FOOSC in the revised manuscript. Given the considerable numbers of isolated pathogens from diseased tobacco, we firstly determined whether the strains belonging to *Fusarium oxysporum* species complex (FOOSC) through morphological and phylogenetic analysis of *tef1* genes. Only the strains were determined belonging to FOOSC, we could selected the *tef1*, *tub2*, *cmdA*, and *rpb2* genes rather than other genes to distinguish the different species of FOOSC (preliminary findings are omitted from this manuscript).

Let us explain to you the reason for this description. *Fusarium oxysporum* species complex is a taxon belonging to a subgenus that has taxonomic status below the genus *Fusarium*, it is above 30 species such as *F. oxysporum*, *F. cugenangense*, *F. odoratissimum*, *F. foetens*, *F. tardichlamydosporum*. At first, the identification of the species mainly relied on morphological differences, but with the development of molecular biology, it is possible to distinguish morphologically similar but genetically different species by using different genes, Therefore, in recent years, "*Fusarium oxysporum*" has evolved into "*Fusarium oxysporum* species complex", and new species have been continuously identified through molecular biology.

The haplotype subspecific classification system was introduced by Chang et al. (2006) and later expanded upon by O'Donnell et al. (2008, 2009) to include strains from both the FOOSC and *Neocosmospora* (formerly the *F. solani* (FSSC) species complex). This classification system is based on unique multi-locus genotypes within the species complex, aimed to resolve communication problems among public health and agricultural scientists (O'Donnell et al. 2008). Chang et al. (2006) proposed a standardised haplotype nomenclature system that depict the species complex, species and genotype. O'Donnell, Lombard et al. (2019) used phylogenetic analysis to establish Epytification of *Fusarium oxysporum* through four genes (*tef1*, *cmdA*, *tub2*, and *rpb2*). This is currently the best method to identify different species in the *Fusarium oxysporum* species complex.

Summary, "320 FOOSC" mentioned in this study refers to the 320 strains belonging to the subspecies of *Fusarium* spp. In the *Fusarium oxysporum* species complex, the specific species are unknown. Whereas "132" indicates that 132 of 320 *Fusarium* spp. strains were identified as species by phylogenetic analysis, and these 132 *Fusarium* spp. were located in a new branch in the phylogenetic tree, so in this study, these 132 *Fusarium* spp. strains were identified as a new species and named *Fusarium tobaccum* sp. nov.

References:

1. Chang DC, Grant GB, O'Donnell K, et al. 2006. Multistate outbreak of *Fusarium keratitis* associated with use of a contact lens solution. *Jama*. 296: 953–963.
2. O'Donnell K, Sutton DA, Fothergill A, et al. 2008. Molecular phylogenetic diversity,

multilocus haplotype nomenclature, and in vitro antifungal resistance within the *Fusarium solani* species complex. *J Clin Microbiol.* 46: 2477–2490.

3. O'Donnell K, Gueidan C, Sink S, et al. 2009. A two-locus DNA sequence database for typing plant and human pathogens within the *Fusarium oxysporum* species complex. *Fungal Genet Biol.* 46: 936–948.
4. Lombard L, Sandoval-Denis M, Lamprecht SC, and Crous PW. 2019. Epitypification of *Fusarium oxysporum*—clearing the taxonomic chaos. *Persoonia.* 43:1-47.

Comment #2. Line 222-224:"The 132 isolates of the FOSC" ... to ... "characteristics.": Depending on phylogenetic analysis there were 7 isolates of *F. tobaccum* previously, but you mentioned 132 isolates identified as "*F. tobaccum*". This represents a big ambiguity, and the authors are using a multi gene joint analysis for 7 isolates only but in conclusions, they mentioned using multi gene for 132.

Response: Thank you for your comments. We are so sorry for this big ambiguity. In view of your comments, we have removed 125 of 132 strains of *Fusarium* isolated from tobacco that identified based on *tef1* gene phylogenetic analysis and related parts in the revised manuscript, and only 7 strains were retained and identified using multi gene phylogenetic analysis and morphology.

Comment #3. In material and method: The authors mentioned about 132 isolates of tobacco belonged to the *F. tobaccum* sp. nov. directly after collection. This identification was depending on what? How could the researchers determine the species and gave a new species directly without any procedure before. In addition, the authors mentioned to FOSC isolates. The table contained on pathogenicity. This is ambiguous. The ideas do not flow logically.

Response: Thank you for your comments. I agree with your opinion, which is more rigorous, and I have revised it in the manuscript (Line 59). The pathogenicity mentioned in Table 1 is not irrelevant in this manuscript. The strains isolated and identified in this study need to be verified by Koch's postulates to determine that they are pathogenic to tobacco. The pathogenicity in the table is the result of Koch's postulates verification.

Comment #4. Morphological for *Fusarium tobaccum*: The authors are depending on characterizations of species not helpful to consider as new species. The authors mentioned characterizations not useful to distinct for a new species from other *Fusarium* spp.

Response: Thank you for your comments. I agree with your opinion, *Fusarium oxysporum* species complex (formerly the *F. oxysporum*) has subtle morphological differences, not only between *F. tobaccum* sp. nov. and *F. cugenangense* as mentioned in this study, it is an objective fact that the morphological differences between other species in FOSC are also very subtle. However, the identification of *F. oxysporum* species complex is mainly based on molecular biology and supplemented by morphology. In other words, without the current identification methods of molecular biology, the taxon of the subgenus *F. oxysporum* species complex would not exist, the taxon of the subgenus complex species was established because of molecular biology.

3. Response to comments/suggestion of the reviewer #5s' comments

The authors of Xie and his/her colleagues reported the identification of a novel *Fusarium* species named *Fusarium tobaccum* sp. nov., from tobacco plants causing tobacco root rot. This novel species was identified based on the classical morphological and molecular phylogenetic analysis using multi-locus sequences. The tasks associated with this manuscript are substantial and very meaningful. However, i have several comments listed as follows:

Comment #1. The identification of *Fusarium* species is one of the most critical issues in fungal taxonomy. From the phylogenetic analysis, it seems reasonable to classify the *Fusarium* fungus in this study as a novel species. However, the morphological identification needs to be strengthened, as only one strain of LH156101 has been used for cultural and morphological characterization. Usually, different strains of *Fusarium* are easily differentiated in physiological phenotypes, thus, phenotypic analysis with multiple strains is warranted.

Response: Thank you for your approval of this study and your valuable comments. For the identification of morphology, only the morphological result of strain LH156101 is presented in the manuscript. In this study, all the seven strains identified by multi-gene phylogenetic analysis were morphologically characterized and we found that the morphology of seven strains were almost the same. Therefore, only the morphological phenotypes of strain LH156101 were presented, and represented the common characteristics of *F. tobaccum* sp. nov.

Comment #2. In the phylogenetic analysis, since only the *tef1* single gene was used, i think it is arbitrarily to classify the other 125 strains as *Fusarium tobaccum* sp. nov., and that the topological structure of the monogenic phylogenetic tree was different from that of polygenic phylogenetic tree. In other words, even if the *tef1* sequence is exactly the same, it can not be classified as a specific cryptic species under the FOOSC. This can be observed by BLASTn analysis on the *Fusarium*-ID database using individual *tef1* sequence.

Response: Thank you for your comments. I wholeheartedly agree with your viewpoint. In view of reviewer 3's comments and your comments, we have removed 125 of 132 strains of *Fusarium* isolated from tobacco that identified based on *tef1* gene phylogenetic analysis and related content in the revised manuscript, and only 7 strains retained and were identified using multi gene phylogenetic analysis and morphology.

Comment #3. According to the reference Lombard et al., 2019 and the phylogenetic analysis in this study, *F. callistephi*, *F. elaeidis*, and *F. cugenangense* are probably the most closely related species of *Fusarium tobaccum* sp. nov., thus, if possible, comparison of morphological or other physiological and biochemical characteristics or genetic background of these four species would make the results of this paper more rigorous.

Response: Thank you for your comments. I wholeheartedly agree with your viewpoint. We have added morphological contrasts of four closed species to the manuscript. Because of *F. callistephi* and *F. elaeidis* were not isolated in this experiment, the morphological description of these two strains was referred to Lombard et al., 2019. (Line 312-320)

Comment #4. For pathogenicity assays, the authors should evaluate the pathogenicity differentiation according to the disease index, not just the disease grades. The authors also

did not account for whether the disease symptoms was consistent across all plants in each treatment. In figure 5, the symptoms of the four strains looks similar, but they were defined in different disease grades of 2 and 3. The symptoms of vascular discoloration should be exhibited.

Response: Thank you for your comments. The method used in this study to measure the severity of disease by mean disease grade was referenced from Clark et al., 1998. The onset symptoms presented in Figure 4 are only representative, while the average disease level in Table 1 is the result of 10 biological replicates, so there might be discrepancies between the two due to these differences. Your understanding in this regard would be greatly appreciated.

Comment #5. In Line 276, the sporulation quantity was inconsistent with that in Table 6, nor was it the same in L286.

Response: Thank you for your comments. We are so sorry for those errors. We have corrected them in the manuscript.

Comment #6. Since this disease is named "Fusarium wilt of tobacco (FWT)", this should be unified in the text.

Response: Thank you for your comments, which have been corrected in the manuscript. (Line 8, 20, 34, 40, and 50)

Response to the editors' and the reviewers' comments

Dear editors and reviewers,

We deeply appreciate the valuable comments and suggestions provided by you and the reviewers for our manuscript entitled "Identification and biological characteristics of *Fusarium tobaccum* sp. nov., a novel species causing tobacco root rot in Jilin Province, China" (Spectrum 00925-24). We have carefully revised the manuscript and provided a point-by-point response to the review's comments. The modifications according to the reviewers' suggestions are marked in red.

Finally, we believe the modifications have significantly improved the manuscript and thank you very much for your attention to our work. We hope we have addressed all your concerns, and please let us know if there are any further questions.

Sincerely yours,

Jie Gao, Ph. D & Professor
Jilin Agricultural University
jiegao115@126.com

Response to the comments from the Reviewer #1:

The authors answered the comments but still some comments, as following:

1. Abstract: Must mention to all morphological and molecular characteristics for new species that found in results to useful for describing this species

Response: Thank you for your valuable suggestion. We have added the morphological and molecular characteristics for new species that found in results in Abstract.

2. Line 55-57: "These findings" to end: Delete it

Response: Thanks for your suggestion. We have deleted this sentence in the manuscript.

3. CONCLUSIONS: You must add what are you conclude from your study without repeating to results

Response: Thank you for your valuable suggestion. We have added what we conclude from our study and revised this conclusions section carefully.

Response to the comments from the Reviewer #2:

The manuscript entitled "Identification and biological characteristics of *Fusarium tobaccum* sp. nov., a novel species causing tobacco root rot in Jilin Province, China" (Manuscript Number: Spectrum00925-24R1) have been reviewed. The revised

manuscript have modified the problems previously, and satisfied all my earlier concerns.

The innovative findings were supported the experimental data from this work.

Response: Thank you for your recognition for our research.

Re: Spectrum00925-24R2 (Identification and biological characteristics of *Fusarium tobaccum* sp. nov., a novel species causing tobacco root rot in Jilin Province, China)

Dear Prof. Jie Gao:

Your manuscript has been accepted, and I am forwarding it to the ASM production staff for publication. Your paper will first be checked to make sure all elements meet the technical requirements. ASM staff will contact you if anything needs to be revised before copyediting and production can begin. Otherwise, you will be notified when your proofs are ready to be viewed.

Sincerely,
Jonathan Snow
Editor
Microbiology Spectrum